# SAM-CP: Marrying SAM with Composable Prompts for Versatile Segmentation

**Pengfei Chen**[1,2*]  **Lingxi Xie**[2]  **Xinyue Huo**[2]  **Xuehui Yu**[1]  **Xiaopeng Zhang**[2]
**Yingfei Sun**[1]  **Zhenjun Han**[1†]  **Qi Tian**[2]
[1] University of Chinese Academy of Sciences    [2] Huawei Inc.

## Abstract

The Segment Anything model (SAM) has shown a generalized ability to group image pixels into patches, but applying it to semantic-aware segmentation still faces major challenges. This paper presents SAM-CP, a simple approach that establishes two types of **c**omposable **p**rompts beyond SAM and composes them for versatile segmentation. Specifically, given a set of classes (in texts) and a set of SAM patches, the Type-I prompt judges whether a SAM patch aligns with a text label, and the Type-II prompt judges whether two SAM patches with the same text label also belong to the same instance. To decrease the complexity in dealing with a large number of semantic classes and patches, we establish a unified framework that calculates the affinity between (semantic and instance) queries and SAM patches, and then merges patches with high affinity to the query. Experiments show that SAM-CP achieves semantic, instance, and panoptic segmentation in both open and closed domains. In particular, it achieves the state-of-the-art performance in open-vocabulary segmentation. Our research offers a novel and generalized methodology for equipping vision foundation models like SAM with multi-grained semantic perception abilities. Codes are released on `github.com/ucas-vg/SAM-CP`.

## 1 Introduction

The past decade has witnessed a rapid development of vision-aware foundation models Radford et al. (2021); He et al. (2022); Wang et al. (2023c; 2021); Liu et al. (2021). These models apply to a series of visual recognition tasks and serve as a building block for multimodal (*e.g.*, vision-language) understanding. Recently, a powerful foundation model named the *Segment Anything* model (SAM) Kirillov et al. (2023) has attracted a lot of attention. Pre-trained on a large corpus of images, SAM shows an impressive ability to group image pixels into patches and generalizes across various vision domains (*e.g.*, medical images He et al. (2023); Ma et al. (2024); Hu & Li (2023); Roy et al. (2023), camouflaged images Chen et al. (2023b); Tang et al. (2023b), thermal images Chen & Bai (2023), *etc.*) as well as different downstream scenarios, (*e.g.*, image editing Yu et al. (2023c); Xie et al. (2023), 3D recognition Shvets et al. (2024); Wang et al. (2023b); Cen et al. (2023), object tracking Cheng et al. (2023); Yang et al. (2023), *etc.*).

Despite its success, there still exist major challenges in applying SAM to semantic-aware segmentation tasks including semantic, instance, or panoptic segmentation. We notice two lines of research in this direction. The first line (*e.g.*, Grounded-SAM Ren et al. (2024)) heavily relied on a standalone model (*e.g.*, DINO Zhang et al. (2023a) or Grounding-DINO Liu et al. (2024)) to generate proposals and SAM was only used for refinement. This weakens the function of SAM as a foundation model. The second line (*e.g.*, SSAM Chen et al. (2023a), Semantic-SAM Li et al. (2023a), SAM-CLIP Wang & Vasu (2023)) tried to assign a semantic label to each patch produced by SAM. However, in many scenarios, SAM may over-segment an instance into sub-patches, making it difficult to determine which patches belong to the same instance.

This paper presents a novel approach named **SAM-CP** where 'CP' stands for composable prompts. Different from the existing methods, we establish two types of prompts beyond the patches produced by SAM. The idea is illustrated in Figure 1. When a semantic class (in text) is given, the model

---

*This work was done when he was an intern at Huawei Inc, chenpengfei20@mails.ucas.ac.cn
†Corresponding Author, hanzhj@ucas.ac.cn

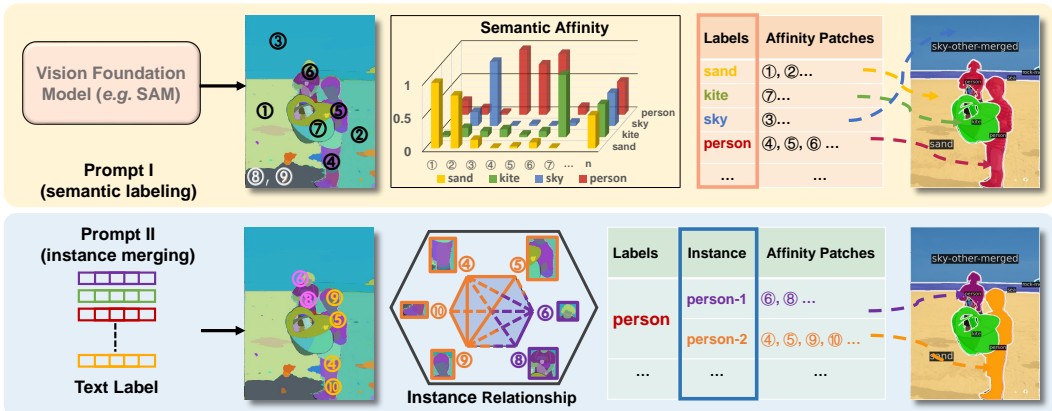

Figure 1: An illustration of how SAM-CP works at the idea level. Given an image and the patches produced by SAM, we first execute Prompt I to find the patches corresponding to any text label (in either closed or open domains), and then, if necessary, execute Prompt II to group the patches within each class into instances. In the upper part, the height of each bar corresponds to the probability that a patch belongs to a text label (yellow, green, blue and red for 'sand', 'kite', 'sky' and 'person'); in the lower part, two patches are connected by a solid line if they belong to the same instance (purple for 'person-1' and orange for 'person-2'). *This figure is best viewed in color.*

needs to determine: **(i) Prompt I:** whether a SAM patch aligns with the text label, and **(ii) Prompt II:** whether two patches belong to the same instance of the corresponding category. Once the model learns to process these two prompts, one can apply a simple traversal algorithm over the SAM patches for semantic segmentation (based on the first prompt) and instance segmentation (adding the second prompt), and further compose them for panoptic segmentation.

A naive implementation of SAM-CP suffers super-linear computational complexity in enumerating class-patch pairs (Prompt I) and patch-patch pairs (Prompt II). To accelerate it, we establish a unified affinity framework, as shown in Figure 2. It involves a query-based mechanism where two types of queries (semantic and instance) are established, and the features extracted from SAM patches are taken as keys. Both the queries and keys are fed into a vision transformer; throughout forward propagation, each query merges the keys with high affinity with it. At the end of affinity propagation, the keys contained in each query form the desired entity (a semantic region or an instance). This implementation allows us to carry out the training procedure efficiently in GPUs.

We train SAM-CP on COCO Lin & Maire (2014) and ADE20K Zhou et al. (2017) and evaluate it on these two datasets as well as Cityscapes Cordts et al. (2016) for both open-vocabulary and closed-domain segmentation. Since SAM-CP is trained to understand text labels, it can easily adapt to unseen classes with a CLIP Radford et al. (2021) text encoder. Extensive experiments demonstrate SAM-CP's ability to cover semantic, instance, and panoptic segmentation with a single model. In particular, it reports state-of-the-art accuracy in open-vocabulary segmentation. Qualitative studies demonstrate that SAM-CP improves the semantic discriminativity of SAM's features. Our research offers a new methodology to equip vision foundation models (*e.g.*, SAM) with a solid and flexible ability for semantic recognition. We expect the proposed approach to gain stronger and more versatile abilities in the future, with the vision foundation models being upgraded and becoming more robust.

## 2 RELATED WORK

In recent years, both the CV and NLP communities have witnessed a rapid development of foundation models. In particular, the vision foundation models have largely evolved from being simply pre-trained for image classification Dosovitskiy et al. (2021); Liu et al. (2021); Wang et al. (2021) to incorporating multimodal information Radford et al. (2021); Fang et al. (2023); Li et al. (2023c); Alayrac et al. (2022) and/or being pre-trained to deal with different tasks Huang et al. (2023); Yu et al. (2023b); Wang et al. (2023c). They improve the accuracy of various downstream visual recognition tasks including detection, segmentation, *etc*.

Recently, SAM Kirillov et al. (2023) appeared as a foundation model for versatile segmentation. Pre-trained on a large corpus with billions of instances, SAM can segment an image into a set of basic patches without tuning. An important advantage of SAM lies in its ability to recognize images

in different domains, yet a disadvantage lies in the lack of semantic labels on each patch. The community has been trying to adapt SAM to various scenarios, including adapting it on other image data (*e.g.*, medical images He et al. (2023); Ma et al. (2024); Hu & Li (2023); Roy et al. (2023), camouflaged images Chen et al. (2023b); Tang et al. (2023b), remote sensing images Chen et al. (2024a); Zhang et al. (2023c); Wang et al. (2023a), *etc.*), raising it to segment 3D objects Shvets et al. (2024); Wang et al. (2023b); Cen et al. (2023), and using it as the pre-processing step of image editing Yu et al. (2023c); Xie et al. (2023). Among these efforts, one of the most challenging topics is to assign the SAM patches with semantic labels. Existing works involved calling other foundation models (*e.g.*, CLIP Radford et al. (2021)) for image tagging Chen et al. (2023a); Li et al. (2023a); Wang & Vasu (2023); Wei et al. (2024), applying SAM as a refinement stage after other detection and/or segmentation models Ren et al. (2024), and generating other variants. However, in many scenarios, SAM may over-segment basic semantic units into sub-patches, which increases our burden of segmentation for specific purposes.

This paper focuses on establishing basic prompts beyond the segmentation results of SAM for versatile segmentation. This is related to a series of query-based algorithms for visual recognition, such as DETR Carion et al. (2020) and the subsequent variants Zhu et al. (2021); Li et al. (2022a); Liu et al. (2022); Zhang et al. (2023a); Liu et al. (2024). Meanwhile, we compute the affinity to determine the relationship between semantic units (*e.g.*, queries and objects), which is related to a few prior work Ahn & Kwak (2018); Yu et al. (2020); Liu et al. (2017) that tried to compute the affinity between pixels and objects to perform segmentation. The idea was also inspired by ViRReq Tang et al. (2023a), a recent work that proposed decomposing complex visual recognition tasks into elementary units to ease annotation and optimization. Further, motivated by the previous open-domain panoptic segmentation methods Yu et al. (2023a); Xu et al. (2023a); Chen et al. (2023c), we leverage a CLIP-based classifier to equip the model with an open-domain recognition ability.

## 3 OUR APPROACH

### 3.1 OVERVIEW: COMPOSITE PROMPTS FOR SEGMENTATION

The overall design of our approach is illustrated in Figure 1. The core idea is to establish two types of prompts beyond SAM Kirillov et al. (2023), a recent vision foundation model that extracts patches from an input image as potential instances. By composing the output of different prompts, the SAM patches are labelled and/or combined into semantic regions and/or instances, and thus versatile segmentation tasks can be performed. We name the approach **SAM-CP**, where 'CP' stands for composable prompts.

Mathematically, let the input image be $\mathbf{I}$. SAM extracts a number of patches, $\mathcal{P} = \{\mathbf{P}_1, \mathbf{P}_2, \dots, \mathbf{P}_N\}$, under segment-everything mode. $N$ is the number of patches and $\mathbf{P}_n$ is the $n$-th patch which is represented as a binary mask of the same shape as the input image. Although SAM is robust across various vision domains, it does not offer a category to each patch and, sometimes, an instance (*e.g.*, a person) may be over-segmented into multiple patches. We design the following two types of prompts.

- **Prompt I – semantic labeling.** Given a text label $\mathbf{T}$ and one patch $\mathbf{P}$, judge if $\mathbf{P}$ can be classified as $\mathbf{T}$.
- **Prompt II – instance merging.** Given a text label $\mathbf{T}$ and two patches $\mathbf{P}_1$ and $\mathbf{P}_2$ classified as $\mathbf{T}$, judge if $\mathbf{P}_1$ and $\mathbf{P}_2$ belong to the same instance of $\mathbf{T}$.

A wide range of segmentation tasks can be accomplished by composing the above two prompts. Firstly, note that the regular semantic segmentation only involves Prompt I (assigning a label to each patch), and instance segmentation is enabled by adding Prompt II (merging over-segmented patches into an instance). Additionally, the two prompts can be iteratively called when segmentation is required at a finer level, *e.g.*, Prompt I for classifying a region into sub-classes, Prompt II for segmenting an instance into parts, *etc*.

### 3.2 EFFICIENT TRAINING WITH A UNIFIED AFFINITY FRAMEWORK

A straightforward implementation of SAM-CP involves executing Prompt I for each patch and then Prompt II for each pair of patches, following which the patches belonging to the same instance can be merged. However, this naive pipeline incurs unsatisfying efficiency because the number of Prompt II to be executed is $O(N^2)$ where $N$ can be hundreds for a regular image. Moreover, the merging procedure requires serial operation and can inevitably encounter conflict in the inference stage (*e.g.*, $\mathbf{P}_1$ and $\mathbf{P}_2$, $\mathbf{P}_1$ and $\mathbf{P}_3$ are considered to be the same instance but $\mathbf{P}_2$ and $\mathbf{P}_3$ are not).

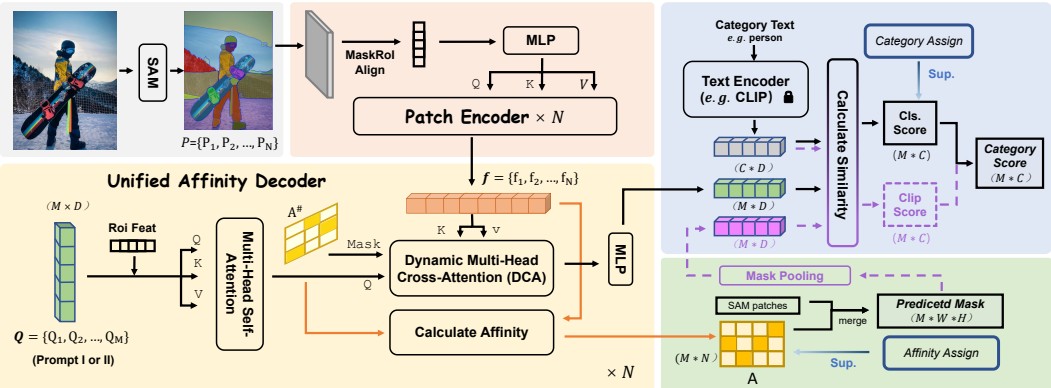

Figure 2: The unified affinity framework as an efficient implementation of SAM-CP. The input image with SAM patches is fed into a patch encoder. Type-I and Type-II prompts appear as two sets of queries. Affinity values are computed and the SAM patches are merged according to the affinity values. Semantic and instance level supervision are added to the merged patches. The purple arrows are present only in the inference stage of open-vocabulary segmentation. *Best viewed in color.*

To accelerate the procedure, we design an equivalent but more efficient mechanism named the **unified affinity framework**. We initialize a set of queries for potential units (*i.e.*, semantic regions and instances) and set all SAM patches as keys. We perform affinity propagation between the queries and keys, gradually merging units with high-affinity scores into a larger unit. With this modified mechanism, the over-segmented patches are merged *on the fly* and no further post-processing is required (*i.e.*, at the end of affinity propagation, the survived units naturally form the output). We illustrate the procedure in Figure 2 and elaborate on the individual modules as follows.

### 3.2.1 PATCH ENCODER

We first extract visual features from the patches. For each patch $\mathbf{P}_n$, we apply a regular backbone (*e.g.*, ResNet50 He et al. (2016) or Swin-L Liu et al. (2021)) equipped with an RoIAlign He et al. (2017) operator to obtain a basic feature vector $\tilde{\mathbf{f}}_n$. We also design a MaskRoI operator to extract more accurate visual features by masking out the background areas. All these features are propagated through a multi-layer perceptron (MLP) and fed into $\omega$ multi-head self-attention layers, where $\omega$ is set to 6 throughout this paper. The feature vector corresponding to $\mathbf{P}_n$ is denoted as $\mathbf{f}_n$.

### 3.2.2 UNIFIED AFFINITY DECODER

This is the core module that performs affinity propagation and merges patches into super-patches. We will elaborate on three key elements below, namely, (1) a set of semantic and instance queries, (2) the algorithm of affinity propagation, and (3) the label assignment mechanism (Section 3.2.3).

**Queries.** The queries are of a similar form as in DETR Carion et al. (2020). Differently, we establish two types of queries for semantic and instance segmentation, respectively. (1) For each text label $\mathbf{T}_c$ where $c \in \{1, 2, \ldots, C\}$ is the class index, we use the language branch of a vision-language model (*e.g.*, CLIP Radford et al. (2021)) to convert it into a query vector, $\mathbf{e}_c^{\mathrm{S}}$, where the superscript 'S' stands for 'semantic'. (2) We also create $N$ instance queries (*i.e.*, assuming that each patch may correspond to an instance, *abbr.* patch-as-query, PasQ) and initialize them using the visual features and position embeddings of the patches, *i.e.*, $\mathbf{e}_n^{\mathrm{I}} \equiv \mathbf{f}_n$, where the superscript 'I' stands for 'instance'. We denote both types of queries as $\mathbf{Q}_m$, $m = 1, 2, \ldots, M$ where $M$ is the number of queries.

**Affinity.** The affinity is mathematically defined as a matrix $\mathbf{A}$ sized $M \times N$. Each entry of $\mathbf{A}$, $A_{m,n}$, denotes the probability that the patch $\mathbf{P}_n$ belongs to the query $\mathbf{Q}_m$. Initially, we set all entries of $\mathbf{A}$ to 1. Then, in each affinity propagation layer (please see below for details), the query vectors (denoted as $\mathbb{Q}$) and the patch features (denoted as $\mathbb{K}$ and $\mathbb{V}$) are fed into a multi-head cross-attention module to update the query vectors for subsequent classification. There are three key modules here. **(1)** The affinity matrix $\mathbf{A}$ serves as a dynamic mask after binary operation in cross-attention, which we call dynamic cross-attention (DCA), to extract the feature from high-affinity patches. **(2)** We insert a module named affinity refinement (AR) to update the affinity matrix $\mathbf{A}$ using the cosine similarity between $\mathbb{Q}$ and $\mathbb{K}$. **(3)** To enhance the query feature, we apply a Query Enhancement (QE) mechanism to fuse the query's feature with the RoI features of its high-affinity regions. The details of DCA, AR,

and QE are described in Appendix A.1. As shown in the ablation (Section 4.5), all these designs contribute to the segmentation accuracy.

### 3.2.3 LABEL ASSIGNMENT AND SUPERVISION

Each query, regardless of its type (semantic or instance), is expected to occupy a set of (one or more) patches and be assigned a class label. So, two sources of supervision are required, which come from the semantic labels and instance IDs, respectively. Each query is supervised by both signals.

**Semantic-level supervision.** We first build a vision-language classifier upon the semantic queries. Following GLIP Li et al. (2022b), the score $S_{m,c}^{\mathrm{cls}}$ of the $m$-th query at the $c$-th class is determined by $\hat{\mathbf{Q}}_m$ and $\hat{\mathbf{e}}_c$, the quantities produced by linearly normalizing $\mathbf{Q}_m$ and $\mathbf{e}_c$ into $[0,1]$. The classification loss Lin et al. (2017b) $\mathcal{L}_{\mathrm{cls}}$ is computed upon $S_{m,c}$ in Equation 1.

$$S_{m,c}^{\mathrm{cls}} = \frac{1}{s} \cdot \hat{\mathbf{Q}}_m^\top \cdot \hat{\mathbf{e}}_c + b, \quad \mathcal{L}_{\mathrm{cls}} = \frac{1}{M} \sum_{c=1}^{C} \sum_{m=1}^{M} \mathrm{FL}\big(\sigma(S_{m,c}^{\mathrm{cls}}), \mathbb{I}_{[c_m^\star = c]}\big). \tag{1}$$

In $S_{m,c}^{\mathrm{cls}}$, $s$ is a learnable scaling factor, and $b$ is a bias parameter. In $\mathcal{L}_{\mathrm{cls}}$, $\mathrm{FL}(\cdot, \cdot)$ is the focal loss, $\sigma(\cdot)$ is the sigmoid activation function, $c_m^\star$ is the ground-truth class label for the $m$-th query (we explain how to compute $c_m^\star$ in the Appendix A.2), and $\mathbb{I}_{[\cdot]}$ the indicator function which takes 1 if the statement is true and 0 otherwise. Note that, if we only perform Type-I prompts, the category text is not necessary, and we only need to perform a binary prediction for each text query to judge whether the category exists in the image. This mechanism was designed for Type-II prompts. An important technical contribution of our work is to unify Type-I and Type-II prompts into one framework so that the training and inference costs are reduced.

**Instance-level supervision.** At the end of affinity propagation, each instance query corresponds to a binary segmentation mask. Let the ground truth contain $K$ instances. We first establish a matching matrix $\mathbf{G}$ (sized $K \times N$) by computing the box-level IoP (intersection-over-patch) and mask-level IoP values between each pair of predicted and true instances; they are considered matched if both IoP values are greater than a pre-defined hyper-parameter $\tau$, *i.e.*, $G_{k,n} = \mathbb{I}_{[\min(\mathrm{IoP}_{\mathrm{box}}, \mathrm{IoP}_{\mathrm{mask}}) > \tau]}$, where $\tau = 0.8$ throughout this paper. If no patches are assigned to an object, the patches with an IoU value of at least $0.5$ will be chosen as candidates of low-quality matching. Based on the matching matrix $\mathbf{G}$, we compute the ground-truth affinity matrix $\mathbf{B}$ (sized $M \times N$, same as $\mathbf{A}$). For each $m$, we first determine if any ground-truth instance (indexed $k_m$) matches the $m$-th query (see the next part for details). If yes, $\mathbf{B}_m = \mathbf{G}_{k_m}$ (*i.e.*, the $k_m$-th row of $\mathbf{G}$ is copied to the $m$-th row of $\mathbf{B}$); otherwise, $\mathbf{B}_m \equiv \mathbf{0}$. Then, following Cheng et al. (2022); Li et al. (2023b), we compute the mask focal loss $\mathcal{L}_{\mathrm{mfl}}$ and the Dice loss $\mathcal{L}_{\mathrm{dice}}$:

$$\mathcal{L}_{\mathrm{mfl}} = \frac{1}{M^*} \sum_{m=1}^{M} \frac{\varepsilon_m}{\max(|\mathbf{B}_m|_0, 1)} \cdot \sum_{n=1}^{N} \mathrm{FL}(A_{m,n}, B_{m,n}), \quad \mathcal{L}_{\mathrm{dice}} = \frac{1}{M^*} \sum_{m=1}^{M} \varepsilon_m \cdot \mathrm{Dice}(\mathbf{A}_m, \mathbf{B}_m), \tag{2}$$

where $\mathrm{FL}(\cdot, \cdot)$ and $\mathrm{Dice}(\cdot, \cdot)$ are the focal loss and Dice loss, respectively, $M^*$ is the number of positive query embeddings in the image, $|\mathbf{B}_m|_0$ is the number of non-zero entries in $\mathbf{B}_m$ (*i.e.*, the number of patches that are assigned to the $m$-th query), and $\varepsilon_m \in \{0, 1\}$ is a weight indicating whether to take the $m$-th query into consideration.

**Determining $\mathbf{B}_m$ and $\varepsilon_m$ for each query.** This procedure is different between the types of queries.

- For a semantic (**Type-I**) query, $\mathbf{Q}_c$ ($c \in \{1, 2, \ldots, C\}$), we first check if the $c$-th class appears in the image. If not, we have $\mathbf{B}_c \equiv \mathbf{0}$ and $\varepsilon_c = 0$. Otherwise, the $c$-th class (as a unique semantic region) must appear in the ground-truth 'instance' set; let the index be $k_c$, thus $\mathbf{B}_c = \mathbf{G}_{k_c}$. We set $\varepsilon_m$ to 1 for the positive embeddings and 0 otherwise.

- For an instance (**Type-II**) queries, we follow the DETR series Carion et al. (2020); Zhu et al. (2021); Zhang et al. (2023a) to apply the Hungarian algorithm to find the best matches between the queries and the ground-truth instances. Differently, we compute the matching cost using more metrics, including the classification loss (cls), the mask focal loss (mfl), the Dice loss (dice), the bounding-box cost (bbox), and the gIoU cost (giou). We will show in experiments that all these components contribute to better segmentation results. After the matching is done, we obtain the index $k_m$ for the $m$-th query ($k_m$ can be null, in which situation the query is ignored), and assign $\mathbf{B}_m = \mathbf{G}_{k_m}$. We set $\varepsilon_m$ to 1 for the positive embeddings and 0 otherwise.

| Method | Backbone | COCO→ADE20K | | | | | ADE20K→COCO | | | | | COCO→Cityscapes | | |
|---|---|---|---|---|---|---|---|---|---|---|---|---|---|---|
| | | PQ | SQ | RQ | AP | mIoU | PQ | SQ | RQ | AP | mIoU | AP | PQ | mIoU |
| MaskCLIP Ding et al. (2023) | VIT-L | 15.1 | 70.5 | 19.2 | 6.0 | 23.7 | – | – | – | – | – | – | – | – |
| FreeSeg Qin et al. (2023) | VIT-B | 16.3 | 71.8 | 21.6 | 6.5 | 24.6 | 21.7 | 72.0 | 21.6 | 6.6 | 21.7 | – | – | – |
| OPSNet Chen et al. (2023c) | VIT-L | 19.0 | 52.4 | 23.0 | – | – | – | – | – | – | – | – | 41.5 | – |
| OpenSeeD Zhang et al. (2023b) | Focal-L | 19.7 | – | – | 15.0 | 29.0 | – | – | – | – | – | – | 41.4 | – |
| HIPIE Wang et al. (2023d) | VIT-H | 22.9 | – | – | 19.0 | 29.0 | – | – | – | – | – | – | – | – |
| X-Decoder Zou et al. (2023) | Focal-L | 21.8 | – | – | 13.1 | 29.6 | – | – | – | – | – | 24.9 | 38.1 | 52.0 |
| MaskQCLIP Xu et al. (2023b) | VIT-L | 23.3 | – | – | – | 30.4 | – | – | – | – | – | – | – | – |
| ODISE Xu et al. (2023a) | VIT-H | 23.3 | 74.4 | 27.9 | 13.0 | 29.2 | 25.0 | 79.4 | 30.4 | – | – | – | 23.9 | – |
| CLIPSelf Wu et al. (2024) | VIT-L | 23.7 | – | – | 13.6 | 30.1 | – | – | – | – | – | – | – | – |
| FrozenSeg Chen et al. (2024b) | CN-L | 25.9 | – | – | 16.4 | 34.4 | – | – | – | – | – | 28.4 | 45.8 | 56.8 |
| FCCLIP Yu et al. (2023a) | CN-L | 26.8 | 71.5 | 32.3 | 16.8 | 34.1 | 27.0 | 78.0 | 32.9 | – | – | 26.8 | 44.0 | 56.2 |
| SAM-CP | CN-L | 27.2 | 77.7 | 32.9 | 17.0 | 31.8 | 28.6 | 78.4 | 34.5 | 21.9 | 34.3 | 29.3 | 41.0 | 47.9 |

Table 1: Accuracy (%) of Open-vocabulary panoptic segmentation (in PQ, SQ and RQ), instance segmentation (in AP) and semantic segmentation (in mIoU). CN-L means ConvNext-L.

| Method | Backbone | Seg. Style | COCO | | | | | ADE20K | | | | |
|---|---|---|---|---|---|---|---|---|---|---|---|---|
| | | | Epoch | PQ | $AP^{det}$ | AP | mIoU | Epoch | PQ | $AP^{det}$ | AP | mIoU |
| DETR Carion et al. (2020) | R50 | reg.+seg. | 50+25e | – | – | 31.1 | – | – | – | – | – | – |
| MaskFormer Cheng et al. (2021) | R50 | seg. | 300e | 46.5 | – | 33.9 | 57.8 | 128e | 34.7 | – | – | – |
| Mask2Former Cheng et al. (2022) | R50 | seg. | 50e | 51.5 | – | 41.7 | 61.7 | 128e | 39.7 | – | 26.4 | 47.7 |
| Mask2Former Cheng et al. (2022) | Swin-L | seg. | 100e | 57.8 | – | 48.6 | 67.4 | 128e | 48.1 | – | 34.2 | 56.1 |
| Mask DINO Li et al. (2023b) | R50 | reg.+seg. | 50e | 53.0 | 48.8 | 44.3 | 60.6 | – | – | – | – | – |
| Mask DINO Li et al. (2023b) | Swin-L | reg.+seg. | 50e | 58.3 | 56.2 | 50.6 | 67.3 | 128e | – | – | – | 56.6 |
| X-Decoder Zou et al. (2023) | Focal-T | seg. | 50e | 52.6 | – | 41.3 | 62.4 | 128e | 41.6 | – | 27.7 | 51.0 |
| X-Decoder Zou et al. (2023) | Focal-L | seg. | 50e | 56.9 | – | 46.7 | 67.5 | 128e | 49.6 | – | 35.8 | 58.1 |
| SAM-CP | R50 | SAM* | 36e | 48.6 | 46.1 | 41.7 | 55.6 | 128e | 38.5 | 28.7 | 25.1 | 42.4 |
| SAM-CP | Swin-L | SAM* | 36e | 52.7 | 50.4 | 45.2 | 61.8 | 128e | 44.4 | 34.6 | 30.3 | 49.4 |

Table 2: Accuracy (%) of panoptic segmentation (in PQ), instance segmentation (in AP), and semantic segmentation (in mIoU) on the COCO-Panoptic and ADE20K datasets. The segmentation (seg.) style 'SAM*' means that we use fixed segmentation results of SAM without any refinement including regression ('reg.') or segmentation ('seg.') refinement.

**The overall loss function.** The overall loss is defined as $\mathcal{L}_{all} = \lambda_{cls} \cdot \mathcal{L}_{cls} + \lambda_{mfl} \cdot \mathcal{L}_{mfl} + \lambda_{dice} \cdot \mathcal{L}_{dice}$, where the loss coefficients are $\lambda_{cls} = 2$, $\lambda_{mfl} = 1$, and $\lambda_{dice} = 1$. We apply the denoising strategy in DINO Zhang et al. (2023a) to improve the training performance.

## 4 EXPERIMENTS

### 4.1 INFERENCE

The inference procedure varies slightly between closed-set and open-vocabulary segmentation. In a **closed domain**, the logit $\mathbf{S}^{cls}$ in Equation 1 is used for classification. The normalized and quantized affinity matrix $\mathbf{A}$ is used for patch merging. For semantic segmentation, we refer to the rows corresponding to $C$ classes and merge all patches surpassing a pre-defined threshold. For instance segmentation, we look up the non-empty instance rows of $\mathbf{A}$, each of which corresponds to an instance. Panoptic segmentation is achieved by combining semantic and instance segmentation results. In an **open domain**, we complement the logit matrix $\mathbf{S}^{cls}$ (sized $M \times C$) with a CLIP-based logit matrix, $\mathbf{S}^{CLIP}$. The remaining part is the same as in a closed domain. To calculate $\mathbf{S}^{CLIP}$, we follow FC-CLIP Yu et al. (2023a) to extract CLIP features using mask pooling on the predicted masks. Then, the feature $\mathbf{S}^{CLIP}$ is obtained by calculating the similarity between the CLIP feature and $\hat{\mathbf{e}}_c$. The final class score is computed as $\mathbf{S}^{ov} = (\sigma(\mathbf{S}^{cls}))^{(1-\kappa)} + (\text{softmax}(\mathbf{S}^{CLIP}))^{\kappa}$, where $\kappa = 0.4$ is a coefficient balancing the closed-domain and open-vocabulary class scores.

**Datasets and evaluation metrics.** We train SAM-CP on the COCO-Panoptic Lin & Maire (2014) and ADE20K Zhou et al. (2017) datasets, and evaluate the models on either closed-domain or open-vocabulary segmentation (with cross-dataset validation and Cityscapes used as test data). COCO-Panoptic (the 2017 version) has 118K training and 5K validation images with 80 'thing' and 53 'stuff' categories. We report the instance segmentation results with the standard AP metric on the 80 'thing' categories. For semantic segmentation, we report the mIoU for all $(80 + 53)$ categories. For panoptic segmentation, the PQ metric is computed for all categories and the 'thing' and 'stuff' subsets individually. ADE20K contains 20,210 images. We use the 150 most common object categories, including 100 'thing' and 50 'stuff' categories. Cityscapes is a street-view dataset with 8 'thing' and 11 'stuff' categories. We inherit the same metric from COCO to ADE20K and Cityscapes datasets.

### 4.2 SETTINGS

**Implementation details.** The proposal masks are generated by SAM with 48 grid points along each axis of the input image. To verify our idea better, we use SAM with VIT-H to generate better patches.

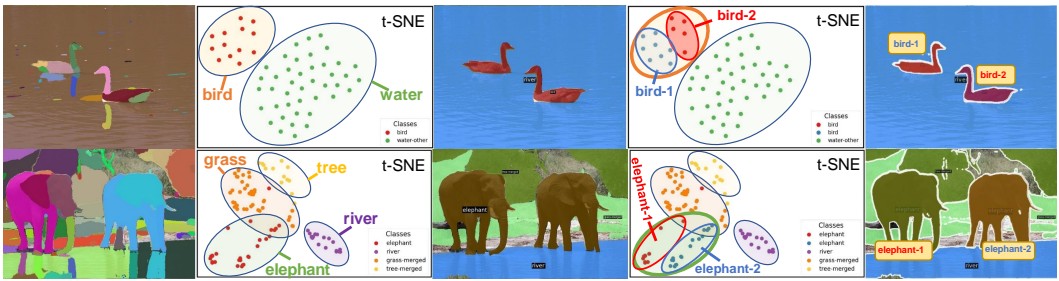

Figure 3: A qualitative study of how SAM-CP works. Each row displays an example. The leftmost column shows the input image with SAM patches; the middle and right parts show the semantic and instance segmentation results, respectively. We use the t-SNE algorithm to project the learned visual features (by SAM-CP; please refer to Figure 4 for the difference from the features of SAM) in a 2D coordinate system. The points with the same color belong to the same semantic class or the same instance (according to the ground truth). *This figure is best viewed in color.*

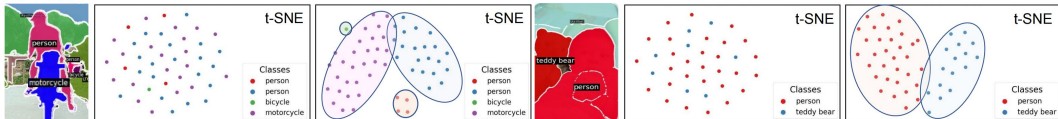

Figure 4: The t-SNE visualization upon the visual features of SAM and SAM-CP. Due to the limited space, only semantic segmentation results are displayed. The points with the same color belong to the same semantic class (according to the ground truth). *This figure is best viewed in color.*

For open-vocabulary segmentation, we use a frozen CLIP image encoder (for a fair comparison, we use the same architecture (ConvNext-L) as the previous best method, FCCLIP Yu et al. (2023a)) as the backbone and equip it with FPN Lin et al. (2017a). For closed-domain segmentation, we establish SAM-CP on ResNet50 (R50) He et al. (2016) and Swin-L Liu et al. (2021). We use the implementation of the MMDetection Chen et al. (2019) (v3.0) library. We use 8 Tesla-V100 GPUs (4/2 images per GPU) for open-vocabulary/closed-domain experiments. The data augmentation strategy follows the DETR series. An AdamW optimizer Loshchilov & Hutter (2019) with a basic learning rate of 0.0002. See Appendix B for further details.

## 4.3 QUANTITATIVE RESULTS

**Open-vocabulary segmentation.** Results are summarized in Table 1. In both **COCO→ADE20K** and **ADE20K→COCO**, SAM-CP surpasses FCCLIP Yu et al. (2023a) (non-SAM-based) and FrozenSeg Chen et al. (2024b) (SAM-based), the previous state-of-the-art methods, in terms of PQ, SQ, RQ for panoptic segmentation, and AP for instance segmentation. In particular, SAM-CP not only reports competitive SQ (for high-quality segmentation), but also achieves a better tradeoff between PQ and RQ; in **COCO→Cityscapes**, SAM-CP achieves a 40.6% PQ, a 29.3% AP and a 47.5% mIoU for versatile segmentation, which brings the best Instance Segmentation performance. We owe the excellent results to the efficient mechanism that combines SAM and CLIP, two open-world foundation models for open-vocabulary segmentation.

**Closed-domain segmentation.** Results are summarized in Table 2. In **COCO-Panoptic**, with the ResNet-50 backbone, SAM-CP achieves a 48.6% PQ, 41.7% AP, and 55.6% mIoU; with a stronger backbone, Swin-L, SAM-CP reports higher segmentation accuracy, with a 52.7% PQ, 45.2% AP, and 61.7% mIoU. In **ADE20K**, SAM-CP achieves a 38.5% PQ, a 25.1% AP, and a 42.4% mIoU on ResNet-50, and a 44.4% PQ, a 30.3% AP, and a 49.4% mIoU on Swin-L. An interesting comparison comes from MaskFormer Cheng et al. (2021) where SAM-CP reports higher PQ and AP but a lower mIoU, which implies its advantageous performance in instance-level recognition. These numbers demonstrate the effectiveness of our methodology, *i.e.*, establishing composable prompts beyond vision foundation models. We will delve into the limitation of SAM-CP in Section 4.6 and explain why this novel mechanism falls short in closed-domain segmentation.

**Summary.** As a new methodology for versatile segmentation, SAM-CP shows a unified pipeline and promising performance over three popular benchmarks. In particular, SAM-CP demonstrates state-of-the-art performance in the open domain. We look forward to the future when stronger foundation models are available and further boost the accuracy of SAM-CP.

| Loss | Label Assignment | Closed-domain (COCO) | | | | Open-domain (COCO→ADE20K) | | | | |
|---|---|---|---|---|---|---|---|---|---|---|
| | | PQ | $AP^{det}$ | AP | mIoU | PQ | SQ | RQ | AP | mIoU |
| all | all | 47.0 | 45.8 | 41.4 | 54.2 | 27.2 | 77.7 | 32.9 | 17.0 | 31.8 |
| w/o $\mathcal{L}_{mfl}$ | w/o mfl | 0.0 | 3.5 | 0.0 | 0.0 | 0.6 | 22.0 | 0.9 | 0.0 | 3.4 |
| w/o $\mathcal{L}_{dice}$ | w/o dice | 41.3 | 35.1 | 34.3 | 48.3 | 23.8 | 73.4 | 29.1 | 15.8 | 28.6 |
| all | w/o mfl | 42.8 | 44.0 | 39.8 | 51.4 | 26.5 | 78.2 | 32.3 | 17.2 | 31.6 |
| all | w/o dice | 45.3 | 44.8 | 40.6 | 53.7 | 26.6 | 76.6 | 32.4 | 16.7 | 31.5 |
| all | w/o box & giou | 45.5 | 44.0 | 40.7 | 53.9 | 25.9 | 76.1 | 31.6 | 16.4 | 30.5 |

Table 3: Accuracy (%) in open and closed domains with different loss terms and matching strategies.

| DCA | AR | MaskRoI | QE | BG | Closed-domain (COCO) | | | | Open-domain (COCO→ADE20K) | | | | |
|---|---|---|---|---|---|---|---|---|---|---|---|---|---|
| | | | | | PQ | $AP^{det}$ | AP | mIoU | PQ | SQ | RQ | AP | mIoU |
| | ✓ | ✓ | ✓ | ✓ | 45.4 | 45.6 | 41.1 | 51.8 | 26.6 | 76.9 | 32.5 | 16.6 | 31.7 |
| ✓ | | ✓ | ✓ | ✓ | 43.5 | 44.0 | 39.9 | 51.1 | 25.8 | 76.8 | 31.3 | 16.3 | 30.5 |
| ✓ | ✓ | | ✓ | ✓ | 44.1 | 45.3 | 40.6 | 51.1 | 25.6 | 74.4 | 31.1 | 16.5 | 30.3 |
| ✓ | ✓ | ✓ | | ✓ | 44.8 | 44.5 | 40.5 | 51.6 | 26.5 | 75.7 | 32.1 | 16.5 | 31.4 |
| ✓ | ✓ | ✓ | ✓ | | 45.2 | 45.4 | 41.3 | 52.6 | 25.5 | 75.7 | 31.2 | 16.1 | 30.3 |
| ✓ | ✓ | ✓ | ✓ | ✓ | 47.0 | 45.8 | 41.4 | 54.2 | 27.2 | 77.7 | 32.9 | 17.0 | 31.8 |

Table 4: Accuracy (%) in open and closed domains with different modules in the SAM-CP framework.

| Strategy | PQ | $AP^{det}$ | AP | mIoU |
|---|---|---|---|---|
| patch-level | 47.0 | 45.8 | 41.4 | 54.2 |
| patch-level, w/o PE | 45.8 | 45.0 | 40.6 | 51.7 |
| image-level | 46.2 | 45.6 | 40.9 | 52.3 |

| Learnable | CLIP | PQ | $PQ^{th}$ | $PQ^{st}$ | AP | mIoU |
|---|---|---|---|---|---|---|
| ✓ | | 17.5 | 15.7 | 21.1 | 11.9 | 19.1 |
| | ✓ | 16.9 | 14.1 | 22.6 | 6.7 | 22.5 |
| ✓ | ✓ | 27.2 | 27.0 | 27.7 | 17.0 | 31.8 |

Table 5: Accuracy (%) on COCO (on R50, 12 epochs) with different strategies of dynamic cross-attention (DCA), where 'patch-level' is the best option, 'PE' denotes the patch encoder.

Table 6: Accuracy (%) on the setting of COCO→ADE20K (on ConvNext-L, 12 epochs) with different classifier types for open domain. 'Learnable' means classifier trained on COCO.

## 4.4 QUALITATIVE STUDIES

We show that SAM-CP learns discriminative visual features beyond SAM. We first show how SAM-CP accomplishes the entire segmentation procedure in Figure 3. From the t-SNE visualization maps, one can see that the features extracted from the SAM patches form clusters that correspond to different semantic classes. Additionally, when instance segmentation is required, the specific cluster can be further partitioned into sub-clusters that correspond to different instances. This aligns with the high-level idea shown in Figure 1, and SAM-CP accomplishes the goal efficiently.

We further compare the visual features learned by SAM and SAM-CP in Figure 4. Not surprisingly, the features extracted by SAM are not semantically discriminative, with samples from different semantic classes overlaying in feature space. Such features are clearly improper for visual recognition purposes. SAM-CP shows much better discriminativity, aligning with the display in Figure 3.

## 4.5 ABLATIVE STUDIES

We study the effectiveness of the design principles and individual modules via ablative studies. In the open vocabulary, we report the results of COCO→ADE20K with a frozen CLIP encoder (ConvNext-L) and a $1\times$ schedule (i.e., 12 epochs). In the closed domain, experiments are performed on the COCO dataset using a ResNet-50 backbone with a $1\times$ schedule.

**Loss and label assignment strategy.** Table 3 shows how different loss functions and label assignment strategies impact the performance. One can see that the mask focal loss is essential, without which the model runs into failure. The dice loss also contributes, especially for instance detection and segmentation. Regarding label assignment, the experiments show clear benefits in introducing more metrics into the weight term to improve the results of bipartite matching.

**Module-level ablation.** There are five components that are helpful to produce better segmentation results, namely, (1) the dynamic cross-attention (DCA) mechanism used for local feature extraction in the decoder, (2) the affinity refinement (AF) strategy by adding a prediction before the sigmoid function in each stage, and (3) the MaskRoI operator which masks out the background region for more accurate visual feature extraction. (4) the query enhancement (QE) which add RoI feature to query embedding. (5) the self-affinity for negative queries to keep the 'segment anyting' ability.

| Proposal types | PQ | AP$^{det}$ | AP | mIoU |
|---|---|---|---|---|
| SAM* | 48.6 | 46.1 | 41.7 | 55.6 |
| SAM* + MD | 51.4 | 51.6 | 45.8 | 57.3 |

Table 7: Accuracy (%) on COCO (with R50, 36 epochs) by adding proposals.

| $\kappa$ | 0.0 | 0.3 | 0.4 | 0.5 | 0.8 | 1.0 |
|---|---|---|---|---|---|---|
| PQ | 17.5 | 26.9 | 27.2 | 26.2 | 22.9 | 16.9 |

Table 8: Accuracy (%) on the setting of COCO→ADE20K with different coefficients, $\kappa$. 0.4 is chosen as the default setting.

| Method | mIoU ↑ | mIoU$_{>0.5}$ ↑ | MR$_{0.25}$ ↓ | MR$_{0.5}$ ↓ | MR$_{0.75}$ ↓ |
|---|---|---|---|---|---|
| Mask DINO | 76.3 | 83.0 | 4.2% | 10.1% | 32.3% |
| SAM | 71.1 | 79.5 | 8.9% | 16.7% | 39.3% |
| SAM+Merging | 73.3 | 81.2 | 9.0% | 15.0% | 33.3% |

Table 9: A comparison between the mIoU and missing rates with respect to different IoUs for COCO (val2017) instance segmentation. Here, mIoU is the IoU between the highest-IoU proposal and the ground truth, and mIoU$_{>0.5}$ means we only calculate the mIoU for the instances that match the best proposal with an IoU higher than 0.5. MR$_x$ denotes the proportion of instances that have no matched proposal with an IoU higher than $x$.

| Method | mAP | $AP_{50}$ | $AP_{75}$ |
|---|---|---|---|
| w.o. general | 13.7 | 30.5 | 10.6 |
| w. general | 13.6 | 30.1 | 10.5 |

Table 10: Accuracy (%) of part segmentation on Pascal Part dataset. The first row shows results of SAM-CP trained only on part datasets, while the second shows those of SAM-CP trained on both part and general datasets.

| Method | SAM - based | SAM / non-SAM time (s/img) | total time (s/img) | PQ | AP | mIoU |
|---|---|---|---|---|---|---|
| FCCLIP Yu et al. (2023a) | no | 0 / 0.24 | 0.24 | 25.2 | 16.4 | 32.6 |
| Frozen Seg Chen et al. (2024b) | yes | 3.41 / 1.07 | 4.48 | 25.9 | 16.4 | 34.4 |
| SAM-CP | yes | 3.41 / 0.21 | 3.62 | 27.2 | 17.0 | 31.8 |
| SAM2-CP | yes | 1.61 / 0.18 | 1.79 | 27.9 | 18.0 | 32.6 |

Table 11: Inference time comparison. 'SAM / non-SAM' means the SAM and non-SAM modules.

Table 4 summarizes the ablation on these five components. Each of them contributes individually and they combined to boost the baseline by at least $1.0\%$ in all the reported metrics.

**The design of DCA.** Among the above modules, DCA requires further investigation. We report the performance of three DCA options, differing from each other in whether cross-attention is computed at the patch level or image level, and whether the patch encoder is used. As shown in Table 5, the patch-level cross-attention with a patch encoder works the best, implying that we can extract sufficient visual features from the SAM's output which often contains hundreds of patches.

**Classifier for open-vocabulary segmentation.** Table 6 shows the impact of classifiers for open-vocabulary segmentation. With a single closed-set (learnable) and CLIP (frozen) classifier, SAM-CP reports a $17.5\%$ PQ and a $16.9\%$ PQ, respectively. After the classifiers are fused, the closed-set and open-vocabulary scores are balanced, resulting in a much higher $27.2\%$ PQ. Interestingly, the closed-set and CLIP classifiers are better at instance and semantic segmentation, respectively, and the fused classifier excels in both scenarios, with a larger gain for instance segmentation.

**Adding extra proposals.** We add the proposals extracted by the pre-trained Mask DINO (MD) model into the pool of candidate patches. Details are provided in Appendix B.1. Table 7 shows improved segmentation results, inspiring us that SAM does not generate ideal patches.

**Average coefficient for open-vocabulary segmentation.** In Table 8, we ablate the average coefficient $\kappa$ defined in Section 4.1 on the setting of COCO→ADE20K. The results show that $0.4$ is the best option in the geometric average between the closed-set and CLIP classification scores.

**Time comparison.** We chose two SOTA methods to compare their time performance within our open-vocabulary setting in Table 11. All experiments were carried out using an RTX 4090. Compared with SAM-based FrozenSeg, our method has less time in non-SAM modules. When replacing SAM with SAM2, the speed becomes faster while the performance is higher.

**Part segmentation.** We conduct part segmentation on the Pascal Part datasets Chen et al. (2014), featuring 4465 images with 93 part categories. Table 10 presents the performance, and Figure 8 offers visualizations. The results' first and second lines stem from models trained solely on part categories and on both part and general ones, respectively. The model demonstrates stable and reliable part - segmentation performance across different segmentation types. SAM relies on low-level cues like color instead of semantics for predictions, as seen in its tendency to separate arm cuffs from skin. Our SAM-based approach inherits these drawbacks, but future tuning on this data may improve results. Additionally, we crop the original image using the instance segmentation map or bounding box, enlarge the sub-image for extra SAM segmentation to obtain more parts, shown in Figure 7, and iteratively use prompt I and prompt II for finer semantic and instance segmentation.

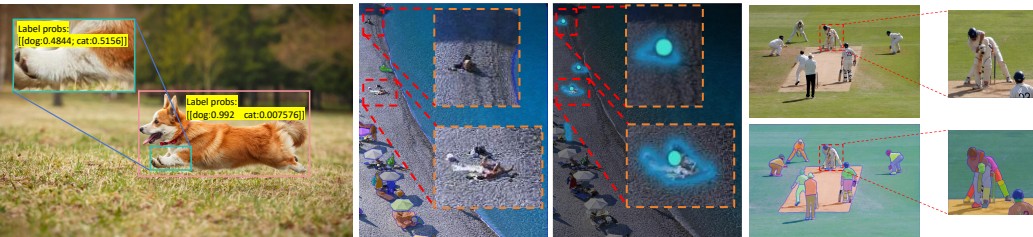

Figure 5: Direct classification for parts and 'parts of the whole'.

Figure 6: Dynamic prompts for looking for small objects.

Figure 7: Interactive SAM calling yields finer results.

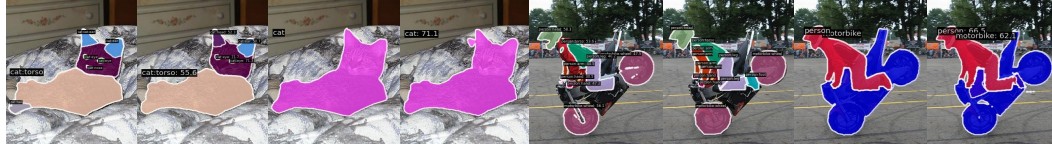

Figure 8: The visualization of part and general instance segmentation. The result is obtained with one model and text labels of different granularities. On the left are the GTs, and on the right are the results.

## 4.6 DISCUSSION

**Advantages.** SAM-CP inherits a clear advantage from SAM in the generalized ability across different visual domains. SAM-CP achieves this by decomposing low-level pixel grouping (offered by SAM) from high-level semantic recognition (offered by the composable prompts). In the meantime, the imperfection of SAM limits the segmentation accuracy of SAM-CP, especially in the closed-set vision benchmarks. We showcase this point in Table 9 where the patches generated by SAM are compared to those generated by Mask DINO Li et al. (2023b), a state-of-the-art segmentation model. We find that SAM suffers a higher missing rate (*e.g.*, one cannot find a proposal with an IoU larger than $0.5$ for $16.7\%$ instances; the rate is not significantly smaller even if we refer to the ground-truth masks to merge some proposals) than Mask DINO (where the rate is only $10.1\%$).

**Classification mechanism.** SAM-CP does not directly classify each patch but uses a query to conduct cross-attention to all high-affinity patches and then fuses their features as the query feature for classification. This enables the model to use richer and more complete information for semantic classification. Figure 5 shows an example. Three patches are present, corresponding to a dog's torso, tail, and leg. Both 'cat' and 'dog' labels are in the semantic space, but it is difficult for each of the three patches alone to classify itself as 'dog' or 'cat'. The model, instead of assigning a high affinity between a class label ('dog' or 'cat') and any patch, assigns a high affinity between these three patches (note that, no matter what the class is, 'dog' or 'cat', they belong to the same instance). After they have been merged, the visual features are often sufficient for classification ('dog' vs. 'cat'). In the example, the 'dog's leg' patch gets an ambiguous classification score (dog: 0.4844, cat: 0.5156) initially, but when it is merged with the other patches (as a 'dog'), the score becomes distinguishable.

**Limitations.** In other words, SAM failed to find some objects or incorrectly merged two or more objects into one patch, and SAM-CP cannot save such loss. This contributes to the deficit in the segmentation accuracy (*e.g.*, AP or mIoU) compared to Mask DINO. We display some typical examples in Appendix A.3. This issue can be alleviated by dynamically adding denser point prompts to the regions (*e.g.* small objects). We show a case in Figure 6, where small targets were found with this simple mechanism. Additionally, the inference speed of SAM-CP is bound by that of SAM; once a more efficient vision foundation model is available, our framework can be seamlessly transplanted.

## 5 CONCLUSIONS

In this paper, we propose SAM-CP, a novel approach that equips SAM with semantic and instance segmentation abilities. At the core of SAM lies two composable prompts, which determine (1) whether a SAM patch aligns with a text label and (2) whether two SAM patches belong to the same instance, respectively. The idea is implemented using a unified affinity framework for efficient training and inference. We show both quantitative and qualitative results with panoptic segmentation on COCO, ADE20K and Cityscapes for both open-vocabulary and closed-domain segmentation. Our study offers a new methodology to make use of the vision foundation models such as SAM.

## ACKNOWLEDGE

This work was supported in part by the Key Deployment Program of the Chinese Academy of Sciences, China under Grant KGFZD145-23-18 and the Strategic Priority Research Program of Chinese Academy of Sciences under Grant E1XA310103. In addition, we would like to thank Zhaoyang Wei for his technical support during the article - experiment process and the rebuttal phase.

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

# A    DETAILS OF OUR APPROACH

## A.1    AFFINITY COMPUTATION

**MaskRoI.** To extract patch-level features, we use the MaskRoI operator to mask out the background areas. Specifically, we first extract the RoI features using the RoIAlign operator on the minimum outer rectangle $\mathbf{P}_n^{\mathrm{rt}}$ of the patch $\mathbf{P}_n$. Then, we transfer the binary mask by rescaling $\mathbf{P}_n$ into the same size (width and height) as the RoI features (*i.e.*, the down-sampled size of $\mathbf{P}_n^{\mathrm{rt}}$). Finally, the RoI features are multiplied by the region mask to obtain the MaskRoI features.

**Dynamic cross-attention.**    Rather than performing global computation, we use dynamic cross-attention (DCA) to guide the cross-attention operator in focusing on the local features. This is motivated by deformable attention Zhu et al. (2021). The cross-attention operator follows the same structure of multi-head attention Carion et al. (2020), and the matrix $\mathbf{A}$ serves as the dynamic attention mask. We binarize $\mathbf{A}$ by setting the value of each entry to 1 if the value is smaller than the threshold (*e.g.*, 0.5) which means that the area is masked and ignored in computing the cross-attention. As described in Section 3.2.2, the query vectors (as $\mathbb{Q}$) and the patch features (as both $\mathbb{K}$ and $\mathbb{V}$) are fed into a multi-head cross-attention with the dynamic attention mask.

**Affinity refinement.** Affinity refinement (AF) is used to update the affinity matrix $\mathbf{A}$ in a coarse-to-fine manner. We first calculate the similarity $\hat{\mathbf{A}}$ between the query vectors ($\mathbb{Q}$) and patch features ($\mathbb{K}$), which will be described in Algorithm 1. The affinity matrix $\mathbf{A}$ is obtained by conducting an element-wise sigmoid activation $\sigma(\cdot)$ on $\hat{\mathbf{A}}$. For the first stage, the $\hat{\mathbf{A}}$ is obtained by calculating similarity; in the subsequent stages, the $\hat{\mathbf{A}}$ is obtained by stacking the similarity of the previous stages, which is called affinity refinement.

**Query enhancement.** The query enhancement (QE) mechanism aims to fuse the query's features with the RoI features of its high-affinity regions. Each query will predict the affinity values and merge the patches according to the values as the mask prediction. The minimum bounding rectangle of the estimated mask will be used to extract the RoI feature. In order to save GPU memory and speed up calculations, we directly merge the minimum bounding rectangles of the patches. Then, the RoI features and the updated query features (mentioned in Section A.1) are averaged as the new query features for query enhancement.

## A.2    SUPERVISION

**Ground-truth class labels.** Let $c_m^\star$ be the ground-truth class label for the $m$-th query. For a Type-I prompt, the $m$-th query corresponds to the $c_m^\star$-th category label. Hence, the objective of the $m$-th

---

**Algorithm 1** Affinity Similarity Calculation

**Input:** Query vectors $\mathbb{Q}$, Patch features $\mathbb{K}$, Head number $\eta$, Stage number $\omega$.
**Output:** Affinity similarity $\hat{\mathbf{A}}$.
**Note:** $\mathbb{Q} \in \mathbb{R}^{M \times D}$, $\mathbb{K} \in \mathbb{R}^{N \times D}$, where $M$ and $N$ is the number of $\mathbb{Q}$ and $\mathbb{K}$. $D$ is the feature dimension, which is a multiple of $\eta$. $s \in \mathbb{R}^1$, $\mathbf{b}_0 \in \mathbb{R}^D$ and $\mathbf{b}_1 \in \mathbb{R}^D$ are the learnable scaling factor and bias parameters to initialize the score to 0.01 for the focal loss.

1: $\mathbb{Q} \leftarrow fc^{\mathbb{Q}}(\mathbb{Q})$;
2: $\mathbb{K} \leftarrow fc^{\mathbb{K}}(\mathbb{K})$;
3: Reshape $\mathbb{Q}$ to $\mathbb{R}^{M \times \eta \times (D/\eta)}$ and transpose $\mathbb{Q}$ to $\mathbb{R}^{\eta \times M \times (D/\eta)}$;
4: Reshape $\mathbb{K}$ to $\mathbb{R}^{N \times \eta \times (D/\eta)}$ and transpose $\mathbb{K}$ to $\mathbb{R}^{\eta \times (D/\eta) \times N}$;
5: $\hat{\mathbf{A}} \leftarrow \frac{\mathbb{Q}\mathbb{K}^\top}{\sqrt{D/\eta}} \in \mathbb{R}^{\eta \times M \times N}$;
6: $\hat{\mathbf{A}} \leftarrow \mathrm{MLP}(\hat{\mathbf{A}}) \in \mathbb{R}^{1 \times M \times N}$
7: Reshape $\mathbf{A}$ to $\mathbb{R}^{M \times N}$;
8: $\hat{\mathbf{A}} \leftarrow \frac{1}{s} \cdot \hat{\mathbf{A}} + \mathbf{b}$, where $\mathbf{b} = \mathbf{b}_1 \mathbb{K} + \mathbf{b}_0$;
9: $\hat{\mathbf{A}}_\omega \leftarrow \hat{\mathbf{A}}$
10: **if** $\omega > 0$ **then**
11:     $\hat{\mathbf{A}} \leftarrow \hat{\mathbf{A}} + \hat{\mathbf{A}}_{\omega-1}$
12: **end if**

---

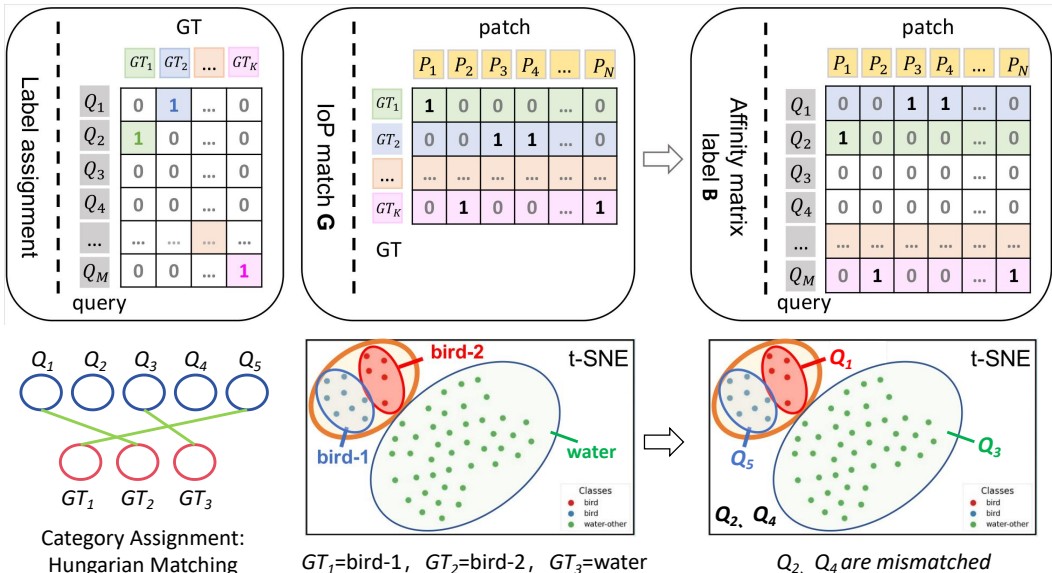

Figure 9: The illustration of how to get the ground-truth affinity matrix **B**. The left is GTs&Q, the middle is GTs&P and the right is Q&P. Line 2 is the t-SNE visualization of category assignment.

query is set to be $c_m^\star$ if the image has the $c_m^\star$-th category label, and 'negative' otherwise. For a Type-II prompt, $c_m^\star$ comes from the label assignment of the Hungarian matching algorithm. The matrix comes from the classification cost, mask focal loss cost, and dice cost. Hence, the objective of the $m$-th query is set to be $c_m^\star$ if it is a positive (matched) query, and 'negative' otherwise.

**Affinity matrix labels.** We use Figure 9 to illustrate how to obtain the affinity matrix label, **B**. The left figure represents the label assignment between the queries and ground truth; the different colors indicate different ground-truth units, where '1' means the query and the ground truth are matched and '0' means not. In the middle figure, **G** means whether each patch is part of the ground-truth unit, where '1' means yes and '0' means no. The colors are consistent with those in the left figure. Based on **G**, we compute the ground-truth affinity matrix **B** (sized $M \times N$, same as **A**), shown in the right figure. For the $m$-th query that matches the $k$-th ground truth in label assignment (left column), the $k$-th row of **G** is copied to the $m$-th row of **B** (with the same color); otherwise, $\mathbf{B}_m \equiv \mathbf{0}$.

The bottom half of Figure 9 shows: (1) The left part is the category assignment processing. We use the Hungarian Matching to achieve one-to-one matching between queries and GTs. The matrix comes from the classification cost, mask focal loss cost, and dice cost. (2) The middle part is the t-SNE visualization between GTs *(GT$_1$, GT$_2$, ...)* and patches. After the category assign procedure, some queries are assigned to the GTs while others are not matched. (3) the right part is the t-SNE visualization which shows the relationship between patches and queries *(Q$_1$, Q$_2$, ...)*.

### A.3 EXAMPLES FOR ADVANTAGES AND LIMITATIONS

**Advantages.** We select the objects (labelled with yellow bounding boxes) that do not match any patches with an IoU greater than $0.5$. In Figure 10, we show examples of objects that do not have complete patches from SAM, while SAM-CP can group the parts to the whole as the object representation. The first line shows a case where SAM fails to predict the whole pair of skis because they are separated. SAM-CP can group them into one instance, which aligns with the definition in COCO. In the second line, the motorcycle is divided into many messy units in SAM, but SAM-CP can group them into a whole instance.

**Limitations.** We also give some examples of objects that are missed by SAM (or have a small IoU with SAM patches). We summarize four scenarios in Figure 11: (a) objects that are very small, (b) objects whose colors are similar to the background, (c) objects that are partly occluded, and (d) object which has cluttered components. Situation (a) is the most serious because the grid-based sampling of SAM is insufficient to find all small targets, and sometimes the image resolution is not high enough for

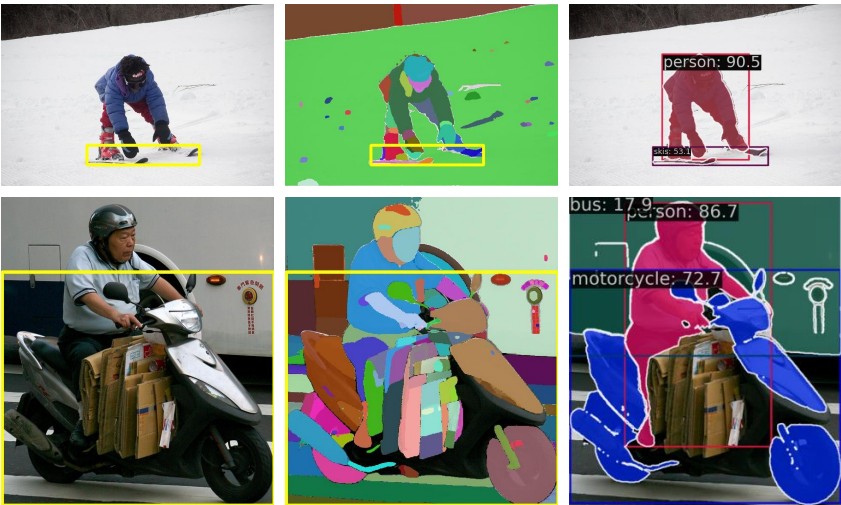

Figure 10: SAM-CP can group the parts to the whole for objects which do not have complete candidate patches. For the first line, SAM segment the pair of ski boards separately. But if our requirement is to treat a pair of skis as one instance, only SAM-CP can put them together. For the second line, in this complex scenario, SAM segments these fragments separately but does not provide a complete motorcycle mask. SAM-CP can stitch these fragments together as fully as possible according to the requirements.

precise prediction. Situation (b) can cause SAM to segment incorrect patches due to indistinguishable low-level patterns (*e.g.*, texture, color, *etc.*). Situation (c) shows that SAM cannot segment well if the object is occluded by other objects, and situation (d) shows that if the object consists of too many cluttered components, SAM cannot segment them very well. This situation limits the quality of SAM patches and, consequently, results in a gap between the accuracy of SAM-CP and the state-of-the-art segmentation methods.

## B DETAILS OF IMPLEMENTATION.

### B.1 SETTINGS FOR COCO.

**Data preparation.** We use the same data augmentation strategy in DETR-based detector Carion et al. (2020); Zhu et al. (2021); Zhang et al. (2023a). We randomly flip the image, resize the image ranging from $1333 \times 480$ to $1333 \times 800$, randomly crop the image, and randomly resize the image again. In the inference stage, we resize the image to $1333 \times 800$ without any other augmentation. For SAM proposal generation, we use the everything mode in SAM with $48$ points per side, a $0.8$ IoU threshold, a $0.9$ stability score threshold, and a $0.7$ NMS threshold.

**Settings for the model.** We use loss weights of $2.0$, $1.0$, and $1.0$ for $\mathcal{L}_{cls}$, $\mathcal{L}_{mfl}$, and $\mathcal{L}_{dice}$, and metric weights of $2.0$, $1.0$, $1.0$, $1.0$, and $1.0$ for the cls, mfl, dice, bbox, and giou terms in the Hungarian matching algorithm. The batch size is 32, where we use 8 Tesla-V100 GPUs (4 images per GPU) for the R50 experiments and 32 GPUs (1 image per GPU) for the Swin-L experiments. The learning rate is set to be $10^{-5}$ at the beginning and is multiplied by $0.1$ after the 8/16/40-th epoch when the total number of epochs is 12/24/50. The threshold $\tau$ in Section 3.2.3 is set to be $0.8$. The number of patch encoders and unified affinity decoders are both set to be 6. The memory usage of each GPU is about 20G for R50 experiments, and training time is about 22 hours for 12 epoch experiments with 8 GPUS. Similar computing resources are needed in further studies.

**Add patches from Mask DINO.** We add the patches from Mask DINO Li et al. (2023b) to show that the upper bound of SAM patches limits our performance. If the quality of patches generated by the foundation model gets better in the future, the segmentation accuracy of SAM-CP will further increase. We saved the segmentation results of Mask DINO and deleted the label result and confidence score. Then, the segmentation results are viewed as the extra proposal and added to the SAM patches.

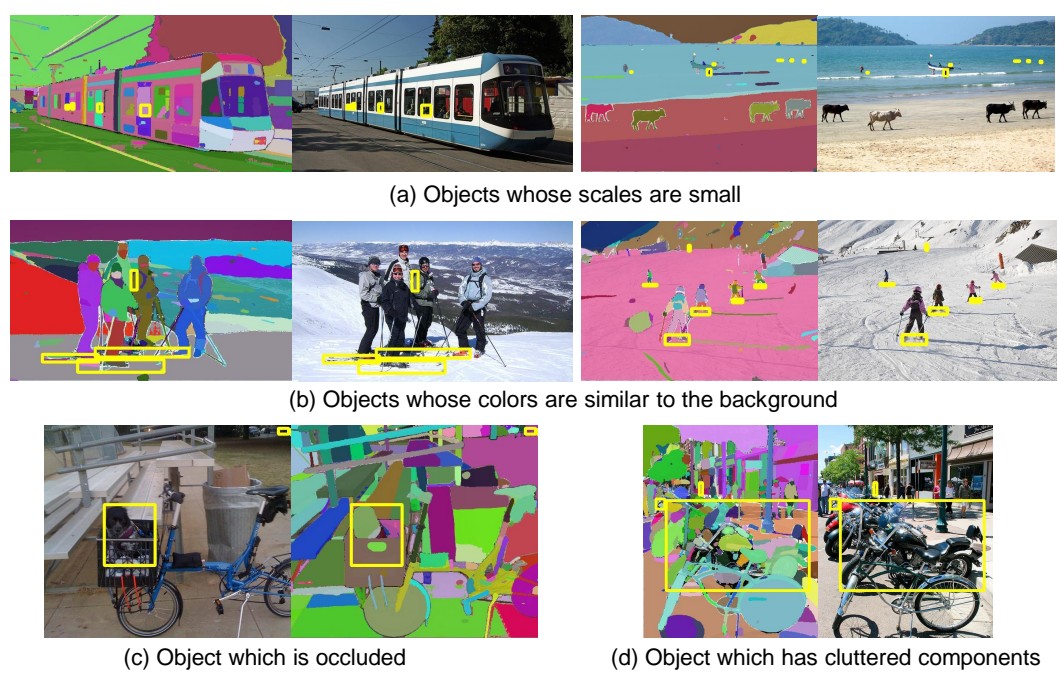

(a) Objects whose scales are small

(b) Objects whose colors are similar to the background

(c) Object which is occluded          (d) Object which has cluttered components

Figure 11: The examples for the limitation of SAM patches. We have summarized the above four situations, and SAM itself has not segmented the target. Our method's premise assumption is to trust SAM's segmentation quality, so the above issues with SAM will limit our final performance. These limitations are also worth further investigation.

## B.2 SETTINGS FOR ADE20K.

**Data preparation.** We use the same data augmentation strategy in Mask2former Cheng et al. (2022) in ADE20K dataset. The image is randomly flipped and resized in the range of $[320, 1280]$. Then, the image is randomly cropped by $2560 \times 640$. In the inference stage, the image is resized to $2560 \times 640$. The hyper-parameters for SAM patch generation are the same as those for the COCO dataset.

**Setting for the model.** We use loss weights of $4.0$, $1.0$, and $1.0$ for $\mathcal{L}_{\text{cls}}$, $\mathcal{L}_{\text{mfl}}$, and $\mathcal{L}_{\text{dice}}$, and metric weights of $4.0$, $1.0$, $1.0$, $1.0$, and $1.0$ for the cls, mfl, dice, bbox, and giou terms in the Hungarian matching algorithm. The batch size is 16 with a basic learning rate of $2 \times 10^{-4}$. The total number of iterations is 160K, and the learning rate will be multiplied by $0.1$ after 135K iterations. Other settings remain unchanged as in the COCO experiments. The memory usage of each GPU is about 20G for R50 experiments, and training time is about 55 hours for 160K iterations experiments (128 epochs) with 8 GPUS.

## B.3 SETTINGS FOR OPEN-VOCABULARY SEGMENTATION.

**Data preparation.** The image pre-processing operations for training and inference on COCO and ADE20K are the same as the closed-domain experiments. When inferencing on Cityscapes, the images are resized to $2048 \times 1024$. When generating SAM patches, because most of the images in Cityscapes are $2048 \times 1024$, it is not suitable to resize the image to a square, so we crop the images on the long side and then feed them into SAM.

**Setting for the model.** The training hyper-parameters are the same as those in close-domain experiments. Only the trainable backbone is replaced by a frozen CLIP image encoder. In the inference stage, the parameter $\kappa$ described in Section 4.1 is set as 0.4. In addition, we follow FCCLIP Yu et al. (2023a) to use the prompt engineering when inference. Each label can derive multiple entries, and we chose the highest score as the classification score for this label.

| $\tau$ | PQ | AP$^{\text{det}}$ | AP | mIoU |
|---|---|---|---|---|
| 0.9 | 43.2 | 42.8 | 38.2 | 49.9 |
| 0.8 | 45.0 | 45.4 | 40.7 | 52.6 |
| 0.7 | 43.3 | 43.3 | 39.9 | 50.3 |
| 0.5 | 42.0 | 34.3 | 37.2 | 48.9 |
| 0.8 (w/o IoP$_{\text{box}}$) | 42.1 | 31.7 | 35.4 | 48.4 |
| 0.8 (w/o IoP$_{\text{mask}}$) | 39.7 | 42.3 | 32.5 | 44.1 |
| 0.8 (w/low quality match) | 47.0 | 45.8 | 41.4 | 54.2 |

Table 12: Accuracy (%) on COCO (with R50, 12 epochs) with different definitions of parts in the training stage.

## B.4 CODES AND DATASETS.

We provide our codes in code.zip file in the Supplementary Material. After the acceptance of our paper, we will make our codes publically available.

The datasets are public datasets; their links are provided here:

- COCO: https://cocodataset.org/
- ADE20K: http://groups.csail.mit.edu/vision/datasets/ADE20K/
- Cityscapes: https://www.cityscapes-dataset.com/

## C MORE EXPERIMENTS

### C.1 MORE ABLATION STUDIES

**The matching threshold and mechanism.** We study the matching threshold and mechanism described in Section 3.2.3, which is key to affinity propagation. We ablate the threshold $\tau$ and the choice of whether to use both the box-level and mask-level IoP in Table 12. We find that $\tau = 0.8$ is a proper threshold. On the other hand, removing either the box-level or mask-level IoP results in a clear accuracy drop in visual recognition, implying the importance of improving the recall of patch merging. In addition, we have added low-quality matching, which will give an object a high IoU patch when it fails to match a positive patch, this will bring performance gain. Therefore, a more accurate mechanism may improve the overall segmentation accuracy, which we leave as future work.

### C.2 MORE COMPARISON

**Comparison with SAM-based method.** Then, we compared our method with OVSAM [1] and prompt segment anything [3] on the COCO dataset to evaluate the instance segmentation ability on Table 13. All other methods need a pre-trained detector on COCO, and the detection results are used as the prompt for SAM. But our localization only relies on the original SAM, without tuning on COCO. The results show our performance is the best:

| Method | Backbone | Detector | mAP | AP50 |
|---|---|---|---|---|
| OVSAM [a] | R50 | FRCNN | 35.8 | 55.6 |
| OVSAM [a] | Swin - B | Detic | 36.7 | 57.2 |
| Prompt segment anything [b] | Res50 | H - Deformable - DETR+tricks | 41.5 | - |
| SAM - CP(ours) | R50 | – | 41.7 | 59.4 |

Table 13: Instance segmentation comparison with SAM-based method.

The comparison between our method and SEEM (Table 14) on the COCO dataset is listed as follows. With the same amount of backbone, we achieved higher performance. We also added a comparison with Semantic SAM (Table 15). Even though it has point prompts in inference, our

method outperforms them by a lot, which fully demonstrates that our method is superior to other SAM-based methods.

| Method | Backbone | PQ | mAP | mIoU |
|---|---|---|---|---|
| SEEM [c] | ViT - L | 52.0 | 43.5 | 60.2 |
| SAM - CP(ours) | Swin - L | 52.7 | 45.2 | 61.8 |

Table 14: Comparison of panoptic segmentation metrics with SEEM.

| Method | Backbone | Prompt(testing) | mIoU |
|---|---|---|---|
| Semantic - SAM [d] | Swin - L | point | 57.0 |
| Semantic - SAM [d] (reproduce) | Swin - L | point | 55.1 |
| SAM - CP(ours) | Swin - L | None | 61.8 |

Table 15: Comparison of semantic segmentation with different SAM-based methods.

By iteratively calling our composable prompts, SAM-CP achieves the instance and semantic segmentation based on the vision foundation model SAM without mask tuning. For SAM2-CP in the first table of this question, we substitute SAM with SAM2 [5] in SAM-CP. SAM2 is faster than SAM. We utilize the CP trained on SAM-CP and transfer the weights to SAM2-CP without re-train, with the performance being 27.9 PQ. This indicates that our method is modular and can be extended to different patch generation methods (whether stronger or faster). Although the current SAM-based method is more time-consuming than the non-SAM-based method, with the improvement of the segmentation accuracy and effectiveness of the SAM-based foundational model, the new bottom-up perception paradigm designed based on its high generalization ability is valuable, which is also one of our main motivations.

[a] Open-Vocabulary SAM. ECCV 2024

[b] Prompt segment anything,(https://github.com/RockeyCoss/Prompt-Segment-Anything)

[c] Segment Everything Everywhere All at Once. NeurIPS 2023

[d] Semantic-SAM: Segment and Recognize Anything at Any Granularity, arXiv:2307.04767

**Traing time comparison.** In addition, SAM-CP only needs 12 epochs of training, while 50 epochs are required for FCCLIP and FrozenSeg.

## D  MORE VISUALIZATION RESULTS

### D.1  VISUALIZATION OF CROSS-DOMAIN EXPERIMENTS

For out-of-domain generalization, we conducted experiments on camouflage object segmentation and provided some visualizations in Figure 12.

### D.2  MORE t-SNE VISUALIZATION

Figure 13 shows more t-SNE visualizations of our approach. The features of patches belong to different semantic/instance groups to different clusters in SAM-CP. Not that, in lines 6–10, the different instances with the same label ('train', 'elephant', 'person', 'horse', and 'umbrella') cluster together to the same semantic area but separate to different instances in the t-SNE feature space.

### D.3  MORE SEGMENTATION EXAMPLES

We show more segmentation results in Figure 14. Columns 1 and 2 are the original images and SAM patches. Columns 3 and 4 are the ground truth and prediction of instance segmentation. Columns 5 and 6 are the ground truth and prediction of panoptic segmentation.

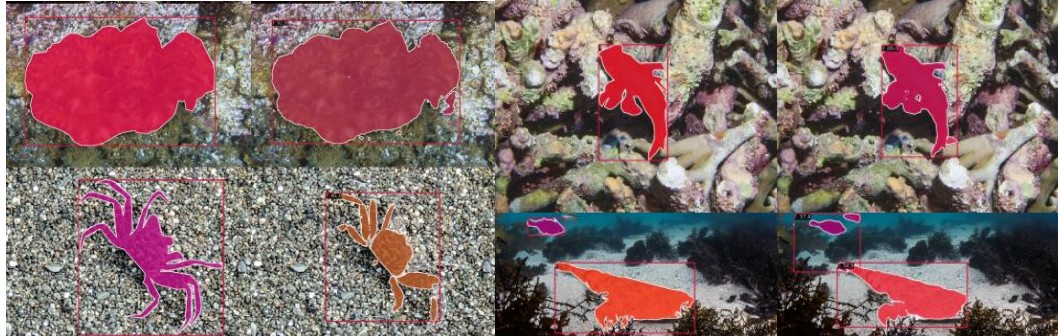

Figure 12: The visualization of camouflage object segmentation on COD10K to show the out-of-domain generalization. Left is the GTs and right is the results.

# E    BROADER IMPACTS AND SAFEGUARDS

**Broader Impacts.** In our view, our research may not have a negative societal impact. Firstly, the comparative methods and datasets used in our research are publicly available and do not make biased decisions that could unfairly impact specific groups. Secondly, our method does not fall under generative models; we aim to enhance the multi-task segmentation capabilities of models. There are no direct pathways for our model to be applied in any negative manner. Therefore, our research does not pose any negative societal impact.

**Safeguards.** We have not released any data or models with a high risk of misuse. The proposed model is trained on benchmark datasets such as COCO, ADE20K, and Cityscapes. These datasets are publicly available, widely used in the field of computer vision, and have undergone comprehensive safety risk assessments. In this paper, we clearly specify the sources of the datasets and code used and provide appropriate links in the references section. Upon completion of the review process, we will make the code and data of this work publicly available to the community. This research does not involve any human subjects experiments or studies, nor does it involve crowdsourced experiments or human subjects research. All experiments were conducted using codes and GPU servers.

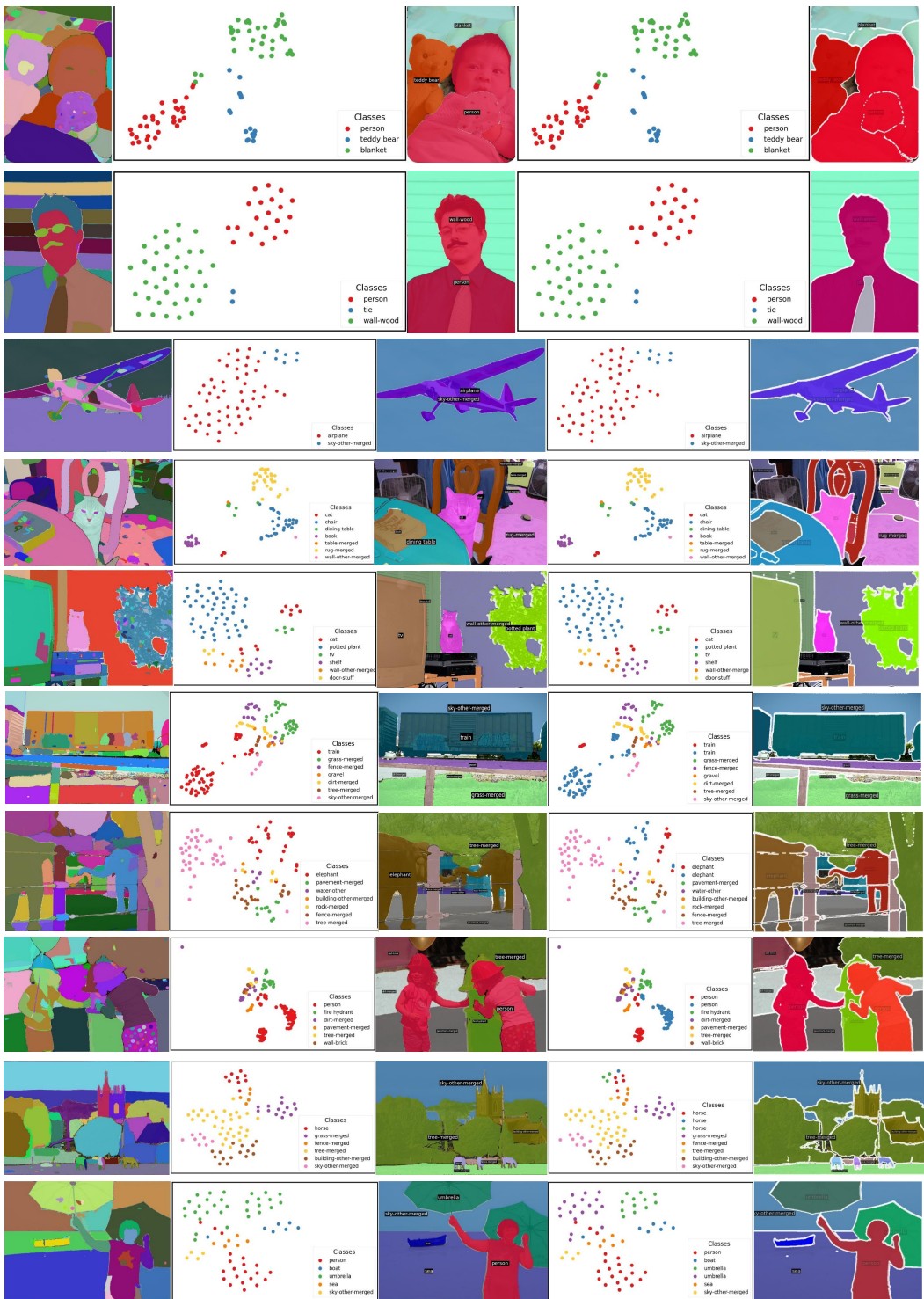

Figure 13: More visualization of a qualitative study on how SAM-CP works.

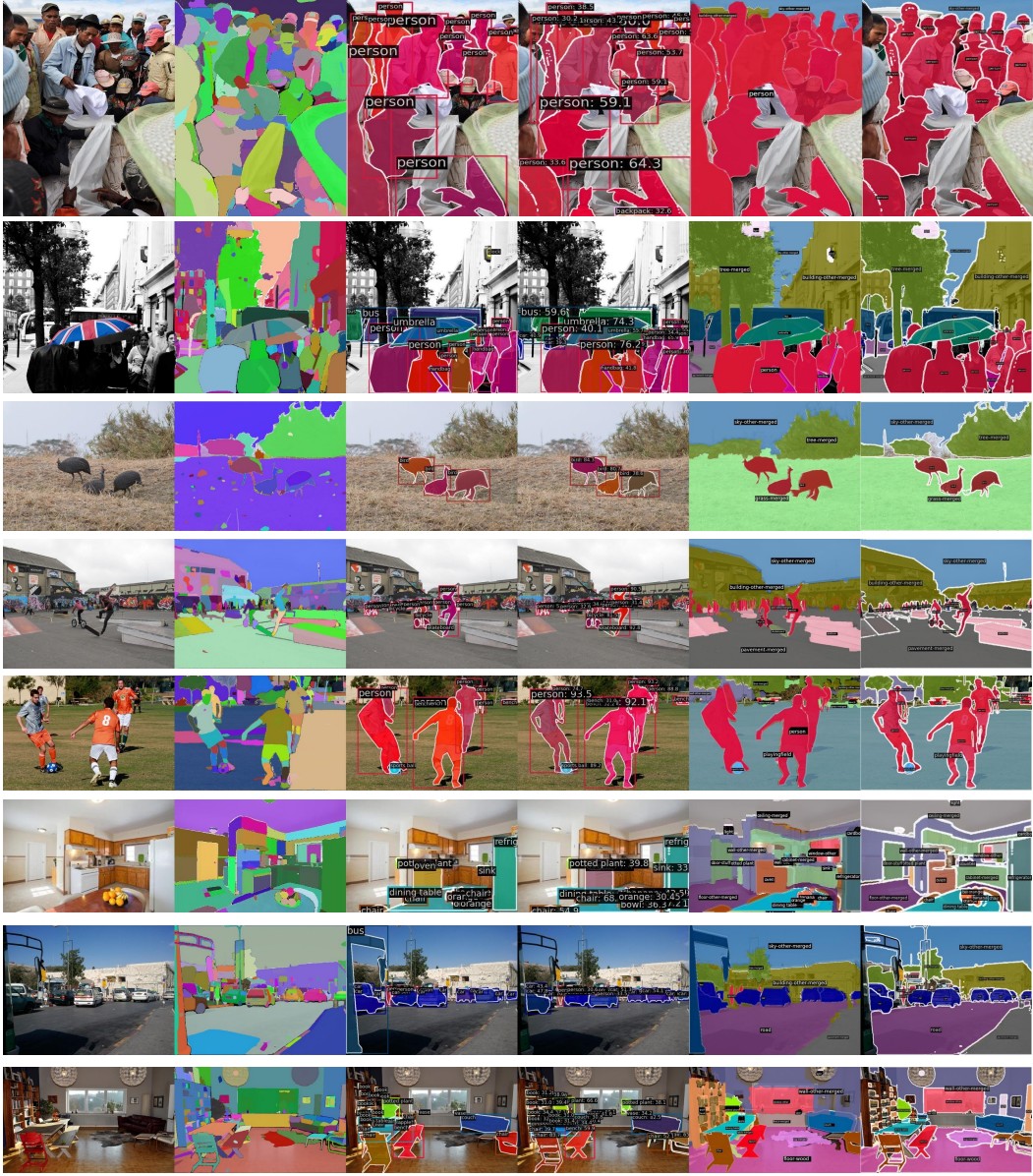

Figure 14: More visualizations of segmentation prediction. Columns 3 and 4 are the ground truth and prediction of instance segmentation. Columns 5 and 6 are the ground truth and prediction of panoptic segmentation.

