# OpenReview forum: "SAM-CP: Marrying SAM with Composable Prompts for Versatile Segmentation"
_ICLR.cc/2025/Conference — ICLR 2025 Poster_

### Official Review · Reviewer_3KqL · 2024-10-27

**Soundness:** 3
**Presentation:** 3
**Contribution:** 2
**Rating:** 6
**Confidence:** 4

**Summary:**

**Objective**

SAM-CP proposes a method to further refine the output of the Segment Anything Model (SAM). The aim is to transform SAM’s original, class-agnostic, variance-scale patches into two more informative segmentation types: Semantic Segmentation and Instance Segmentation.

**Proposed Framework**

The training framework operates as follows:

- Input: The framework takes the initial class-agnostic segments (CA-segments) from SAM, along with category text.
- Processing:
SAM Patches: The SAM patches are first processed through self-attention.
Queries: Queries (which can be target categories in text or instance queries) are processed through self-attention, then used to refine the features by performing cross-attention with the SAM patches.
Affinity Calculation: The resulting  N  patch features and  M  query features are used to calculate affinity for instance grouping.
Semantic Classification: Finally, the learned features are compared with CLIP text input features for semantic classification.

The entire pipeline is trained in a fully supervised, end-to-end manner.

**Results**

Experiments show that this framework achieves state-of-the-art (SOTA) results when trained on one dataset and tested on another.

**Strengths:**

1. I believe this task is important. Transforming the class-agnostic SAM into a usable semantic and instance segmentation is a valuable pursuit. This directly connects to the question of how to leverage the advance of large-scale vision foundation models into practical application.

2. The proposed method demonstrates strong performance on commonly used open-world segmentation frameworks, achieving state-of-the-art (SOTA) results when trained on one dataset and tested on another. This includes benchmarks such as: COCO → ADE20K; ADE20K → COCO; COCO → Cityscapes

3. The overall experinement and results presentation is good.

**Weaknesses:**

# At a higher level:
The advantage of SAM lies not only in its ability to segment an entire image but also in its prompt-based segmentation, which introduces more accurate masks.
I think this method uses SAM only as an initial “superpixel” generator, which somewhat overlook SAM’s full potential.

# Generalization:
The proposed method requires training on specific datasets to learn how to merge patches, which seems to be not very generalizable.
This approach may not translate well from natural images to medical images, or even from natural images with large object segments to tasks requiring finer, more detailed segmentation. Is this the case in this setting. Would like to hear author feedback on this point.

# Splitting and Merging:
The proposed method only focuses on merging, which to some degree simplifies the problem. I believe splitting is equally important as merging, though it seems it is not within the scope of this method.

# Comparison with Other Methods:
Most of the results are compared with pre-SAM methods. I wonder how this method compares with other SAM-based semantic segmentation/instance segmentation methods, such as Grounding SAM and SEEM etc. It would be interesting to see a comparison with other SAM-based methods and discuss the advantage of such approach.

**Questions:**

1. At line 150, the authors state that the two prompts can be iteratively called when finer-level segmentation is required. I’m not entirely sure I understand this; my impression is that granularity is determined by SAM’s results. Could the authors clarify this point?

2. How do the authors view this approach in comparison to SAM-based generalized segmentation? Most other approaches appear to be training-free and straightforward to use. It would be interesting to discuss the advantages that SAM-CP offers.

---

> ### Author Response · Authors · 2024-11-24
> **Response to Reviewer 3KqL (Part # 1)**
>
> **Q1** The advantage of SAM lies not only in its ability to segment an entire image but also in its prompt-based segmentation, which introduces more accurate masks. I think this method uses SAM only as an initial “superpixel” generator, which somewhat overlooks SAM’s full potential.
>
> **A1** A good question! We concur that the strength of SAM is not just confined to its capacity to segment entire images but also extends to its prompt-based segmentation capabilities, which yield more precise masks. However, we do not underestimate SAM's comprehensive potential. Firstly, it is valuable in performing non-prompt-based segmentation, for example, to extract more efficient visual features， which uses grid points to extract all potential patches as many as possible for recognition. Secondly, when our approach offers two additional types of prompts which can be integrated with SAM's prompts for more flexible segmentation -- an interesting example comes from the scenario that an instance is partially occluded and split into patches; in this situation, our Type-II prompt can provide additional clues (whether these patches belong to the same instance) for the top-level model to make the final decision.
>
>
> **Q2** Generalization: The proposed method requires training on specific datasets to learn how to merge patches, which seems to be not very generalizable. This approach may not translate well from natural images to medical images or even from natural images with large object segments to tasks requiring finer, more detailed segmentation. Is this the case in this setting? Would like to hear author feedback on this point.
>
> **A2** A good question! To further show the advantages of our SAM-CP, we conduct experiments on more perspectives. Due to the time limitation, we experiment with part segmentation and out-of-domain generalizations：
>
> **For part segmentation**, we experimented the part segmentation on Pascal Part datasets. Pascal Part has 4465 images with 93 part categories. The performance is listed as follows:
> |Part segmentation |mAP|AP$_{50}$|AP$_{75}$|AP$^{s}$|AP$^{m}$|AP$^{l}$|
> |-|-|-|-|-|-|-|
> |SAM-CP|13.7|30.5|10.6|8.3|21.1|24.2|
>
> We also provide some part segmentation visualization in Figure 16 of the main PDF (rebuttal section, page 24). It's important to note that SAM's predictions are primarily based on low-level information such as color, rather than semantics. For instance, SAM might separate the cuffs of clothing from the skin of the arms, even though the cuffs are still part of the arms. Since our approach builds upon SAM's results, it inherits some of these limitations. In the future, further tuning of SAM for this type of data could lead to more refined part segmentation.
>
> **For out-of-domain generalization**, we conducted experiments on camouflage object segmentation and provided some visualizations in Figure 17 of the main PDF (rebuttal section, page 24).
>
> **Q3** Splitting and Merging:
> The proposed method only focuses on merging, which to some degree simplifies the problem. I believe splitting is equally important as merging, though it seems it is not within the scope of this method.
>
> **A3** A good question! Merging patches with two prompts is the main idea of SAM-CP, but we have also taken the "splitting" into consideration by interactively calling two prompts. With an instance segmentation result, if a human wants to split it for fine-grained segmentation, they can crop the sub-image with the segmentation map and enlarge it for extra SAM everything-mode segmentation. Then, more details can be segmented, as shown in Figure 19 in the main PDF (rebuttal section, page 25). When the foundation models become stronger in the future, these two prompts can be called interactively for fine-grained segmentation.

---

> ### Author Response · Authors · 2024-11-24
> **Response to Reviewer 3KqL (Part # 2)**
>
> **Q4** Comparison with Other Methods:
> Most of the results are compared with pre-SAM methods. I wonder how this method compares with other SAM-based semantic segmentation/instance segmentation methods, such as Grounding SAM and SEEM, etc. It would be interesting to see a comparison with other SAM-based methods and discuss the advantage of such approach.
>
> **A4:** Thanks for the question. We first compare the SOTA SAM-based work Frozen Seg [2] (in our Table I, CVPR2024) with SAM-CP because that work experiments on our same setting (COCO->ADE20k). The result indicates that SAM-CP achieves superior open-vocabulary panoptic segmentation outcomes, which verifies the efficacy of composable prompts. Moreover, it demonstrates that the performance is not solely reliant on the internal capacity of SAM:
>
> |Method|backbone |PQ|AP|mIoU|
> |-|-|-|-|-|
> | Frozen Seg [2] | ConvNext-L | 25.9|16.4|34.4|
> | SAM-CP  | ConvNext-L |27.2|17.0|31.8|
> | SAM2-CP | ConvNext-L |27.9|18.0|32.6|
>
> Then, We compared our method with OVSAM [1] and prompt segment anything [3] on the COCO dataset to evaluate the instance segmentation ability. All other methods need a pre-trained detector on COCO and use the detection results as the prompt for SAM. But our localization only relies on the original SAM, without tuning on COCO. The results show our performance is the best:
>
>
> |Method|Backbone|Detector|mAP|AP50|
> | - | - | - | - | -|
> |OVSAM [1]|R50|FRCNN|35.8|55.6|
> |OVSAM [1]|Swin-b|Detic|36.7|57.2|
> |Prompt segment anything [3]|Res50|H-Deformable-DETR+tricks|41.5|-|
> |SAM-CP(ours)|R50|--|41.7|59.4|
>
> The comparison between our method and SEEM on the COCO dataset is listed as follows. With the same amount of backbone, we achieved higher performance. In addition to comparing with the several methods you mentioned, we also added a comparison with Semantic SAM. Even though it has point prompts in inference, our method outperforms them by a lot, which fully demonstrates that our method is superior to other SAM-based methods:
> |Method|Backbone|PQ|mAP|mIoU|
> | - | - | - | - | - |
> |SEEM [4]|ViT-L|52.0|43.5|60.2|
> |SAM-CP(ours)|Swin-L|52.7|45.2|60.8|
>
> |Method|Backbone|Prompt(testing)|mIoU|
> | - | - | - | - |
> |Semantic-SAM [6]|Swin-L|point|57.0|
> |Semantic-SAM [6] (reproduce)|Swin-L|point|55.1|
> |SAM-CP(ours)|Swin-L|None|61.8|
>
> By iteratively calling our composable prompts, SAM-CP achieves the instance and semantic segmentation based on the vision foundation model SAM without mask tuning. For **SAM2-CP**  in the first table of this question, we substitute SAM with SAM2 [5] in SAM-CP. SAM2 is faster than SAM. We utilize the CP trained on SAM-CP and transfer the weights to SAM2-CP without re-train, with the performance being 27.9 PQ. This indicates that our method is modular and can be extended to different patch generation methods (whether stronger or faster). Although the current SAM-based method is more time-consuming than the non-SAM-based method, with the improvement of the segmentation accuracy and effectiveness of the SAM-based foundational model, the new bottom-up perception paradigm designed based on its high generalization ability is valuable, which is also one of our main motivations.
>
> [1] Open-Vocabulary SAM. ECCV 2024
>
> [2] Frozenseg: Harmonizing frozen foundation models for open-vocabulary segmentation. In CVPR 2024.
>
> [3] Prompt segment anything,(https://github.com/RockeyCoss/Prompt-Segment-Anything)
>
> [4] Segment Everything Everywhere All at Once. NeurIPS 2023
>
> [5] SAM 2: Segment Anything in Images and Videos, arxiv.org/abs/2408.00714.
>
> [6] Semantic-SAM: Segment and Recognize Anything at Any Granularity, arXiv:2307.04767
>
> **Q5** At line 150, the authors state that the two prompts can be iteratively called when finer-level segmentation is required. I’m not entirely sure I understand this; my impression is that granularity is determined by SAM’s results. Could the authors clarify this point?
>
> **A5** A good question! We will clarify that:
>
> * **For the initial segmentation:**
> When the **Prompt I** is called and the category name is "person", SAM-CP can find out all the patches that belong to person and merge them as the semantic segmentation results.
> When the **Prompt II** is called under the category name "person", SAM-CP can distinguish each two pathes whether they belong to the same instance and then the instance segmentation result is obtained.
>
> * **For finer segmentation**：
> In order to segment the part 'Arm', the person in the original image is cropped with the instance segmentation map (or the bounding box). Enlarge the sub-image to conduct extra SAM everything-mode segmentation, and more parts will be obtained.
> Then, the prompt-I and prompt-II are iteratively called to achieve the semantic and instance segmentation of 'Arm'.

---

> ### Author Response · Authors · 2024-11-24
> **Response to Reviewer 3KqL (Part # 3)**
>
> **Q6** How do the authors view this approach in comparison to SAM-based generalized segmentation? Most other approaches appear to be training-free and straightforward to use. It would be interesting to discuss the advantages that SAM-CP offers.
>
> **A6:** A good question! Indeed, there are some training-free methods to implement open vocabulary now, but most of them use SAM for segmentation and CLIP for classification, which can only achieve semantic segmentation. They cannot achieve instance segmentation. Also, the current segmentation quality is also not good enough, so, tuning is also needed for better performance, especailly in vertical field. In our response to Question 4, a detailed comparison was drawn between the advantages of our method and those of other SAM-based approaches. It is worth highlighting that all of these renowned SAM-based methods necessitate training. Specifically, certain methods train prompt generators to furnish prompts to SAM, while others require fine-tuning SAM on extensive datasets. Arguably, these methods cannot be regarded as truly training-free. In sharp contrast, our method attains state-of-the-art (SOTA) performance without the need to fine-tune SAM, which stands as a prominent and distinctive advantage of our approach.
>
> **Moreover**, the majority of these methods are founded on the combination and enhancement of prior techniques. Nevertheless, our method is fundamentally differentiated in its conceptual framework. We embrace a bottom-up strategy, whereby the global information of the target is discerned from local observations. This approach diverges substantially from the conventional up-bottom methods. The triumph of our method implies that this novel concept harbors even more substantial potential for development within the domain of segmentation.

---

> > ### Comment · Reviewer_3KqL · 2024-11-25
> >
> > ## General Comments:
> >
> > My major concern is how useful the proposed method is to the SAM community. I am particularly interested in learning more about the response to A2 and A5. Specifically, after training, can the model be applied across various general-purpose scenarios, with the level of detail (granularity) determined dynamically by the input text promp.
> >
> > ## Individual response:
> >
> > A1: With prompt-based segmentation, I still do not see how the proposed method is useful. The proposed method focuses on merging different mask, and in the prompt-based segmentation, it only returns a small amount of overlapping prompts. This seems outside the scope of the study, and that is fine.
> >
> > A2: Thanks for providing results on part segmentation. Were the part segmentation results obtained with weights trained on other segmentation, or specifically trained on for part segmentation dataset?
> >
> > A3: The proposed figure seems to suggest asking SAM to segment the zoomed-in images again and then use SAM_CP for further segmentation. This approach makes sense when the image has good resolution and the minimal unit for segmentation is still the raw SAM results.
> >
> > A4: Thanks for adding the experiments.
> >
> > A5: Thanks for the clarification. The granularity of the final results can indeed be adjusted using this method. Same as question 2, can general and part segmentation performed by one model and if such model is trained in this study.
> >
> > A6: Thanks for the response.

---

> > > ### Author Response · Authors · 2024-11-27
> > > **Re-rebuttal to Reviewer 3KqL**
> > >
> > > We are pleasure to have cleared your comments on Q2 and Q6. To address the remaining concerns, we provide further response below.
> > >
> > > **About A1**: With prompt-based segmentation, I still do not see how the proposed method is useful. The proposed method focuses on merging different mask, and in the prompt-based segmentation, it only returns a small amount of overlapping prompts. This seems outside the scope of the study, and that is fine.
> > >
> > > **Answer:** To avoid misunderstanding, we would like to clarify that by "prompt-based segmentation" you meant SAM's ability to segment a target with multiple points clicked on the object. If we misunderstood, please feel free to point it out. Below is the response based on our understanding.
> > >
> > > (1) From a general perspective, the studied task is "important" and "valuable" by "transforming the class-agnostic SAM into a usable semantic and instance segmentation" (borrowed from your review). The proposed method works in this direction: once the model is trained, it offers two extra types of *automatic* prompts for versatile segmentation. Of course, we agree that prompt-based segmentation (based on *human* prompts, i.e. telling the algorithm that the clicked masks belong to the same instance) is more accurate but it also introduces additional labeling costs.
> > >
> > > (2) Practically, the Type-II prompts offer complimentary information to instance segmentation, and **it can be used together with human prompts to reduce human labor**. For example, if an object contains a number of basic masks (which means that a labeler needs a few clicks to prompt SAM), when he/she makes the first click and obtains one basic mask, we can call the trained SAM-CP model (the Type-II prompt) to judge if other basic masks and the first one belong to the same instance. The segmentation may finish automatically (by finding all other masks) or at least provide suggestions for the labeler to approve, which can potentially save labeling costs. In addition, if the labeler makes mistakes (i.e. clicking more or fewer masks), the Type-II prompt can detect errors and provide suggestion.
> > >
> > > In summary, we thank the question that helps us to clarify how SAM-CP can be used with human annotation. We look forward to further research in integrating human and automatic prompts towards a more efficient system.
> > >
> > >
> > >
> > >
> > > **About A2&A5**: Thanks for providing results on part segmentation. Were the part segmentation results obtained with weights trained on other segmentation, or specifically trained on for part segmentation dataset? Can general and part segmentation performed by one model and if such model is trained in this study.
> > >
> > > **Answer:** The current part segmentation results were obtained on a small-scale part segmentation dataset (PascalParts) to quickly demonstrate the ability of SAM-CP for part segmentation.
> > >
> > > To show that SAM-CP can do both segmentation with a single model, we combined the general and part segmentation datasets and trained the model again. The new model can perform instance and part segmentation by calling the Type-II prompt multiple times with text labels of of different granularities.
> > >
> > > The following table shows the quantitative result:
> > > |Part segmentation |mAP|AP$_{50}$|AP$_{75}$|AP$^{s}$|AP$^{m}$|AP$^{l}$|
> > > |-|-|-|-|-|-|-|
> > > |SAM-CP w.o. general|13.7|30.5|10.6|8.3|21.1|24.2|
> > > |SAM-CP w. general|13.6|30.1|10.5|8.2|21.3|23.6|
> > >
> > > （1） The results in the table are evaluated on the part segmentation (93 part categories). The first line of results is derived from the model that has been trained exclusively on the part categories, whereas the second line stems from the model that has been trained on both the part and the general categories. The discoveries evidently demonstrate that, irrespective of whether merely part segmentation is executed or an equilibrium between general and fine-grained segmentation is achieved, the model's performance in part segmentation persists to be highly stable and dependable.
> > >
> > > （2） The visualization in Figure 20 of the main PDF (rebuttal section, page 26) shows that with one model, reneral and part categories can be segmented when given text labels of different granularities.（For clarity, we will draw separately.）
> > >
> > >
> > >
> > > **About A3**: The proposed figure seems to suggest asking SAM to segment the zoomed-in images again and then use SAM_CP for further segmentation. This approach makes sense when the image has good resolution and the minimal unit for segmentation is still the raw SAM results.
> > >
> > > **Answer:** Agreed and thanks for the comments. By interactively calling SAM and two prompts, finer segmentation can be achieved. We would also like to add that with zoom-in image and calling SAM again, the minimal unit of SAM results can be finer, which enable SAM-CP to segment more details. When the foundational model gets stronger in the future, the pluggable SAM-CP can have a greater impact.

---

> > > > ### Comment · Reviewer_3KqL · 2024-11-28
> > > >
> > > > Thank you to the authors for the detailed response, which addressed most of my concerns. I would like to revise my rating to 6. My change of opinion is primarily based on the following two points:
> > > >
> > > > 1. In the rebuttal, the authors demonstrated the potential for generalization across different granularities when the model is jointly trained on part-level and general datasets.
> > > > 2. By interactively using two types of prompts in combination **with SAM**, finer segmentation can be achieved.

---

> > > > > ### Author Response · Authors · 2024-11-28
> > > > > **Thank you**
> > > > >
> > > > > We thank the reviewer for recognizing our efforts. We will combine the additional content into the final version.

---

### Official Review · Reviewer_pKRW · 2024-10-28

**Soundness:** 3
**Presentation:** 3
**Contribution:** 3
**Rating:** 6
**Confidence:** 3

**Summary:**

The paper presents SAM-CP that extends SAM by incorporating composable prompts to improve segmentation. SAM-CP utilizes Type-I prompts to align SAM-generated patches with text labels and Type-II prompts to merge patches belonging to the same instance based on the text information. Experimental results show that SAM-CP performs well in open-vocabulary segmentation and shows some improvement in semantic discriminativity across various segmentation types.

**Strengths:**

1. This work presents a method for improving SAM for versatile segmentation.

2. SAM-CP considers segmentation tasks from a novel perspective, introducing two types of composable prompts: Type-I for matching SAM patches with text labels and Type-II for merging patches belonging to the same instance.

3. The dynamic mechanism offers novel design insights for handling cross-attention computations.

4. Generally, the paper is well-written.

5. The figures and tables are informative.

**Weaknesses:**

1. Prompt I classifies each SAM-segmented patch, potentially leading to ambiguities. For example, without contextual information, it’s challenging to determine if a patch representing clothing on a person should be classified as part of the person or as an independent clothing item. The proposed CP lacks a clear strategy to handle the inherent ambiguity in segmentation without surrounding context.

2. The performance gains are not significant, such as mIoU for the COCO->ADE20K and COCO->Cityscapes settings in Table 1.

3. Table 2's comparison may be unfair due to the use of SAM, which is trained on a significantly larger scale. This complicates attributing improvements directly to the proposed method.

4. Although SAM-CP's speed heavily relies on SAM, the paper lacks a clear analysis of the computational efficiency of the proposed non-SAM modules. It is crucial to examine whether these modules actually enhance efficiency.

5. Other Issues:
- Figure 1's many colors and mixed visual elements are hard to read.
- The description of patches needs greater specificity. For example, as vaguely mentioned in Line 139, it's not specified if the patches are segments from SAM's segment-everything mode or just features from SAM's backbone.

**Questions:**

1. I would like to know if SAM-CP has effective strategies for addressing the aforementioned issues of segmentation ambiguity?

2. Although the speed is influenced by SAM's computational efficiency, can you provide a detailed analysis of the speed of non-SAM modules and compare it with other SAM-based methods? Alternatively, could replacing SAM with a lightweight variant like EfficientSAM provide sufficient performance and computational efficiency for comparison with non-SAM methods?

3. Does SAM-CP offer any advantages in training time compared to other SAM-based methods?

4. Since SAM's segmentation results may not always be precise, the Affinity matrix used by the Unified Affinity Decoder could potentially accumulate errors. How is this issue addressed?

---

> ### Author Response · Authors · 2024-11-24
> **Response to Reviewer pKRW (Part # 1)**
>
> **Q1** Prompt I classifies each SAM-segmented patch, potentially leading to ambiguities. For example, without contextual information, it’s challenging to determine if a patch representing clothing on a person should be classified as part of the person or as an independent clothing item.  I would like to know if SAM-CP has effective strategies for addressing the aforementioned issues of segmentation ambiguity?
>
> **A1** Good question! This is exactly why the Type-II prompt was designed in its current form. Note that the Type-II prompt is to judge "whether two patches belong to the same instance **given a specified category label**." In the aforementioned example, whether the two patches (clothing and arm) belong to the same instance depends on the category label -- the answer is yes when the label is "person" but no when the label is "clothing." So, SAM-CP is naturally designed to avoid the ambiguity.
>
> In further detail, here is an illustration of how two types of prompts interact to solve the above case.
>
> * **For a specified category "person"**, when the Type-I prompt is called, both the clothing and arms (of one person) are assigned affinity to this category and classified as "person." Then, for the patches found, the Type-II prompt is called, and the clothing and the left and right arms are considered belonging to the same "person". The "person" instance thus occupies both patches and, if necessary, feeds into a category with finer granularity (see below).
> * **For a specified category "clothing"**, when the Type-I prompt is called, the clothing patch is assigned affinity to this categoty and classified as "clothing", while the left and right arms (of one person) are not. This can achieve the finer granularity classification for those patches belong to "Person".
> * **For a specified category "arm"**, when the Type-I prompt is called, both the left and right arms (of one person) are assigned affinity to this category and classified as "arm," while the clothing patch is not. Then, for the patches classified as 'arm', the Type-II prompt is called, and the left and right arms are considered **not** belonging to the same "arm". Thus, these two patches are separated into two individual instances of "arm."
>
> **Q2** The performance gains are not significant, such as mIoU for the COCO->ADE20K and COCO->Cityscapes settings in Table 1.
>
> **A2:** Thanks for the question. SAM-CP applies a bottom-up mechanism to merge patches, which is inherently friendly to instance segmentation. On the other hand, we find that SAM itself is more robust to instances than semantic regions. In particular, for some large semantic regions (*, e.g.,* sky, mountain, tree, *, etc.*), SAM can suffer from inaccurate boundaries because these regions are not easily prompted. We hope that with the improvement of the segmentation accuracy and effectiveness of the SAM-based foundational model, the new bottom-up perception paradigm designed based on its high generalization ability is valuable, which is also one of our main motivations.
>
> **Q3** Table 2's comparison may be unfair due to the use of SAM, which is trained on a significantly larger scale. This complicates attributing improvements directly to the proposed method.
>
> **A3:** A good question! Really, SAM has undergone extensive training on a significantly larger scale dataset. However, without fine-tuning in closed domain datasets, the quality of SAM's mask is actually inferior to those trained in COCO. The following table compares the mean Intersection over Union (mIoU) and missing rates with respect to different IoUs for COCO (val2017) instance segmentation.
>
> | Method         | mIoU ↑ | mIoU > 0.5 ↑ | MR$_{0.25}$ ↓ | MR$_{0.5}$ ↓ | MR$_{0.75}$ ↓ |
> |----------------|---------|--------------|----------|---------|----------|
> | Mask DINO      | 76.3    | 83.0         | 4.2%     | 10.1%   | 32.3%    |
> | SAM            | 71.1    | 79.5         | 8.9%     | 16.7%   | 39.3%    |
> | SAM+Merging    | 73.3    | 81.2         | 9.0%     | 15.0%   | 33.3%    |
>
> *(Here, mIoU refers to the IoU between the highest-IoU proposal and the ground truth, and mIoU${>0.5}$ indicates that we calculate the mIoU only for instances where the best proposal has an IoU higher than 0.5. MR${x}$ represents the proportion of instances without a matched proposal exceeding an IoU of $x$. )*
>
> The table reveals that SAM patches have a higher missing rate compared to the fine-tuned Mask DINO on COCO. Our SAM-CP, which is based on SAM patches without any fine-tuning, thus faces limitations in close-domain performance. However, when SAM-CP merges patches, it achieves better mIoU and lower missing rates, especially MR$_{0.75}$ (39.3% -> 33.3%). This indicates that after applying composable prompts, SAM-CP delivers improved segmentation outcomes. This also shows that the current performance comes from composable prompts we designed rather than only depending on SAM.

---

> ### Author Response · Authors · 2024-11-24
> **Response to Reviewer pKRW (Part # 2)**
>
> **Q4** Although SAM-CP's speed heavily relies on SAM, the paper lacks a clear analysis of the computational efficiency of the proposed non-SAM modules. It is crucial to examine whether these modules actually enhance efficiency.
> Although the speed is influenced by SAM's computational efficiency, can you provide a detailed analysis of the speed of non-SAM modules and compare it with other SAM-based methods? Alternatively, could replacing SAM with a lightweight variant like EfficientSAM provide sufficient performance and computational efficiency for comparison with non-SAM methods?
>
> **A4** A good question! We opted for two state-of-the-art (SOTA) methods, namely FrozenSeg (which is SAM-based and presented at CVPR 2024) and FCCLIP (a non-SAM-Based method from NeurIPS 2023), to compare their time performance within our open-vocabulary setting. All experiments were carried out using an RTX 4090.
> The outcomes are as follows:
> |Method|SAM-based|backbone |time of SAM (s/img)|time of non-SAM (s/img)|totol time (s/img)|PQ|AP|mIoU|
> |-|-|-|-|-|-|-|-|-|
> | FCCLIP [3]| no | ConvNext-L | -|0.24| 0.24|25.2|16.4|32.6|
> | Frozen Seg [2]| yes | ConvNext-L | 3.41 | 1.07|4.48|25.9|16.4|34.4|
> | SAM-CP | yes | ConvNext-L | 3.41| 0.21| 3.62|27.2|17.0|31.8|
> | SAM2-CP | yes | ConvNext-L | 1.61| 0.18| 1.79|27.9|18.0|32.6|
>
> **FrozenSeg**, being a SAM-based approach, also requires everything-mode segmentation and thus demands the same SAM time as ours. However, our non-SAM module consumes less time and exhibits better performance. **FCCLIP** (the performance here is a reproduced version by both FrozenSeg and us, which is lower than that in the original paper) is a non-SAM-based method, while SAM-CP demonstrates better segmentation performance.
> In **SAM2-CP**, we substitute SAM with SAM2[2] in SAM-CP. SAM2 is faster than SAM. We utilize the CP trained on SAM-CP and transfer the weights to SAM2-CP without re-train, with the performance being 27.9 PQ. This indicates that our method is modular and can be extended to different patch generation methods (whether stronger or faster). Although the current SAM-based method is more time-consuming than the non-SAM-based method, with the improvement of the segmentation accuracy and effectiveness of the SAM-based foundational model, the new bottom-up perception paradigm designed based on its high generalization ability is valuable, which is also one of our main motivations.
>
> [1] SAM 2: Segment Anything in Images and Videos, arxiv.org/abs/2408.00714.
>
> [2] Frozenseg: Harmonizing frozen foundation models for open-vocabulary segmentation. In CVPR 2024.
>
> [3] Convolutions die hard: Open-vocabulary segmentation with single frozen convolutional CLIP. In NeurIPS, 2023.
> [1] SAM 2: Segment Anything in Images and Videos, arxiv.org/abs/2408.00714.
> [2] Frozenseg: Harmonizing frozen
> foundation models for open-vocabulary segmentation. In CVPR 2024.
> [3] Convolutions die hard: Open-vocabulary segmentation with single frozen convolutional CLIP. In NeurIPS, 2023.
>
> **Q5** Figure 1's many colors and mixed visual elements are hard to read.
> The description of patches needs greater specificity. For example, as vaguely mentioned in Line 139, it's not specified if the patches are segments from SAM's segment-everything mode or just features from SAM's backbone.
>
> **A5** Thanks for the kind reminder. We will clarify the colors and mixed visual elements for better understanding. We give a detailed description of "patches" in Line 139：the patches are segments from SAM's segment-everything mode.
>
> **Q6** Does SAM-CP offer any advantages in training time compared to other SAM-based methods?
>
> **A6** A good question! The training time is our advantage. For Table I, we only need 12 epoch to achieve our performance while other methods FCCLIP （non-SAM-based） and FrozenSeg (SAM-based) need 36 epoch. This shows that our approach is better at achieving convergence.
>
> **Q7** Since SAM's segmentation results may not always be precise, the Affinity matrix used by the Unified Affinity Decoder could potentially accumulate errors. How is this issue addressed?
>
> **A7** Thanks for the question. Indeed, our approach can inherit the inaccurate segmentation mask of SAM, but it will not **accumulate** errors (e.g., making errors larger). To reduce the inaccuracy, one can either use a stronger foundation model (e.g., SAM-HQ or SAM2) or apply a post-processing algorithm to refine the boundary. In the Table of **Q4**, the results of SAM2 show the potential， we hope with the improvement of the segmentation accuracy and effectiveness of the SAM-based foundational model, the new bottom-up perception paradigm designed based on its high generalization ability is valuable, which is also one of our main motivations.

---

> > ### Comment · Reviewer_pKRW · 2024-11-25
> >
> > Thank you for addressing my concerns with detailed responses. Most of my concerns have been resolved, and thus I will maintain my initial score.

---

> > > ### Author Response · Authors · 2024-11-27
> > > **Thank you**
> > >
> > > We thank the reviewer for recognizing our efforts. If you still have any questions or interest in our article, please feel free to make further discussion！

---

### Official Review · Reviewer_SuUH · 2024-11-02

**Soundness:** 3
**Presentation:** 3
**Contribution:** 2
**Rating:** 6
**Confidence:** 4

**Summary:**

This paper proposes a framework that combines SAM and CLIP for open-vocabulary universal segmentation.

**Strengths:**

1. The proposed model keeps the zero-shot abilities of CLIP. This is supported by experiments on open-vocabulary segmentations.
2. The ablation studies are sufficient and well explain the design choice of loss functions.
3. This method is easy to follow, with many intuitive figures and visualizations.

**Weaknesses:**

1. The experiments on open-vocabulary segmentations (Table 1) are not well supportive. SAM-CP combines SAM and CLIP but is only comparable to FC-CLIP that is built with CLIP only. Also, the experiments on closed-vocabulary segmentations (Table 2) do not best address the potential of combining SAM and CLIP since it is worse than CLIP-only X-Decoder and supervised SOTA MaskDINO.
2. The advantages of the proposed framework are not well discussed or presented. The performances are only comparable to previous SOTA. This paper should look for more perspectives to highlight its advantages, e.g. efficiency, part segmentation (also see Question 1), few-shot capabilities, out-of-domain generalizations, etc.
3. It could be more convincing if this paper can compare to other SAM-based methods on the same settings and show its advantages. For example, [1] and [2] also work on open-vocabulary segmentation with SAM.
4. As already discussed and illustrated in the paper, SAM-CP fails at small objects since there are no prompts on small objects. The efficiency is limited by SAM.

[1] Open-Vocabulary SAM. ECCV 2024

[2] Segment Everything Everywhere All at Once. NeurIPS 2023

**Questions:**

1. SAM can generate fine-grained masks for part, e.g. head, hand, body. Does SAM-CP have the potential to work on open-vocabulary part segmentation? Quantitative experiments and comparisons to related works like HIPIE would enhance the applicability of this work.

---

> ### Author Response · Authors · 2024-11-24
> **Response to Reviewer SuUH (Part # 1)**
>
> **Q1:** The experiments on open-vocabulary segmentations (Table 1) are not well supportive. SAM-CP combines SAM and CLIP but is only comparable to FC-CLIP that is built with CLIP only. Also, the experiments on closed-vocabulary segmentations (Table 2) do not best address the potential of combining SAM and CLIP since it is worse than CLIP-only X-Decoder and supervised SOTA MaskDINO.
>
> **A1:** Thanks for the question! We respond this question with experiments on **Open domain** and **close domain**:
>
> **For open domain:** In the following table, we present a comparative analysis of two state-of-the-art (SOTA) models, FCCLIP [3] (CLIP-based, presented at NeurIPS 2023) and FrozenSeg [2] (CLIP&SAM-based, presented at CVPR 2024), within the COCO->ADE20K open-vocabulary setting. The initial row reflects the performance metrics reported for FCCLIP in its original publication. The subsequent two rows display the reproduced performance of FCCLIP as reported by the FrozenSeg paper and our own reproduction, respectively. Our method significantly outperforms both FCCLIP and FrozenSeg in terms of results, thereby achieving the SOTA. **In SAM2-CP, we substitute SAM with recently published SAM2 in SAM-CP without re-training.** SAM2 [1] is faster than SAM. We utilize the CP trained on SAM-CP and transfer the weights to SAM2-CP without re-train, with the performance being 27.9PQ. This indicates that our method is modular and can be extended to different patch generation methods (whether stronger or faster). Although the current SAM-based method is more time-consuming than the non-SAM-based method, with the improvement of the segmentation accuracy and effectiveness of the SAM-based foundational model, the new bottom-up perception paradigm designed based on its high generalization ability is valuable, which is also one of our main motivations.
>
> |Method|Backbone|PQ|mAP|mIoU|
> | - | - | - | - | - |
> |FCCLIP [3]|ConvNeXt-L|26.8|16.8|34.1|
> |FCCLIP [3] (frozenseg-reproduce)|ConvNeXt-L|25.1|16.4|32.8|
> |FCCLIP [3] (our-reproduce)|ConvNeXt-L|25.2|16.4|32.6|
> |Frozenseg [2]|ConvNeXt-L|25.9|16.4|34.4|
> |SAM-CP(ours)|ConvNeXt-L|27.2|17.0|31.8|
> |SAM2-CP(ours)|ConvNeXt-L|27.9|18.0|32.8|
>
> [1] SAM 2: Segment Anything in Images and Videos, arxiv.org/abs/2408.00714.
>
> [2] Frozenseg: Harmonizing frozen foundation models for open-vocabulary segmentation. In CVPR 2024.
>
> [3] Convolutions die hard: Open-vocabulary segmentation with single frozen convolutional CLIP. In NeurIPS, 2023.
>
> **For closed domain:** In the realm of open-vocabulary tasks, SAM-CP has achieved state-of-the-art (SOTA) performance. However, in the closed-vocabulary scenario, our SAM-CP exhibited relatively lower performance in closed-set benchmark testing. The following table, which presents statistically analyzed data from the COCO dataset, delves into the reasons behind this. Primarily, this is because we chose not to fine-tune the masks of the visual foundation model SAM, unlike other methods, such as Mask DINO, which were trained on COCO.
>
> | Method         | mIoU ↑ | mIoU > 0.5 ↑ | MR$_{0.25}$ ↓ | MR$_{0.5}$ ↓ | MR$_{0.75}$ ↓ |
> |----------------|---------|--------------|----------|---------|----------|
> | Mask DINO      | 76.3    | 83.0         | 4.2%     | 10.1%   | 32.3%    |
> | SAM            | 71.1    | 79.5         | 8.9%     | 16.7%   | 39.3%    |
> | SAM+Merging    | 73.3    | 81.2         | 9.0%     | 15.0%   | 33.3%    |
>
> *(The table compares the mean Intersection over Union (mIoU) and missing rates with respect to different IoUs for COCO (val2017) instance segmentation. Here, mIoU refers to the IoU between the highest-IoU proposal and the ground truth, and mIoU${>0.5}$ indicates that we calculate the mIoU only for instances where the best proposal has an IoU higher than 0.5. MR${x}$ represents the proportion of instances without a matched proposal exceeding an IoU of $x$. )*
>
> The table reveals that SAM patches have a higher missing rate compared to the fine-tuned Mask DINO on COCO. Our SAM-CP, which is based on SAM patches without any fine-tuning, thus faces limitations in close-domain performance. However, when SAM-CP merges patches, it achieves better mIoU and lower missing rates, especially MR$_{0.75}$ (39.3% -> 33.3%). This indicates that after applying composable prompts, SAM-CP delivers improved segmentation outcomes.

---

> ### Author Response · Authors · 2024-11-24
> **Response to Reviewer SuUH (Part # 2)**
>
> **Q2:** The advantages of the proposed framework are not well discussed or presented. The performances are only comparable to previous SOTA. This paper should look for more perspectives to highlight its advantages, e.g. efficiency, part segmentation (also see Question 1), few-shot capabilities, out-of-domain generalizations, etc.
>
> **A2:** A good question! To further demonstrate the strengths of our SAM-CP, we have conducted experiments from multiple angles. Due to time constraints, we focused on part segmentation and out-of-domain generalization:
>
> **For part segmentation**, we conducted experiments on the Pascal Part dataset, for instance, segmentation tasks. The Pascal Part dataset comprises 4465 images across 93 part categories. The performance metrics are as follows:
> |Part segmentation |mAP|AP$_{50}$|AP$_{75}$|AP$^{s}$|AP$^{m}$|AP$^{l}$|
> |-|-|-|-|-|-|-|
> |SAM-CP|13.7|30.5|10.6|8.3|21.1|24.2|
>
> We also provide some part segmentation visualization in Figure 16 of the main PDF (rebuttal section, page 24). It's important to note that SAM's predictions are primarily based on low-level information, such as color, rather than semantics. For instance, SAM might separate the cuffs of clothing from the skin of the arms, even though the cuffs are still part of the arms. Since our approach builds upon SAM's results, it inherits some of these limitations. In the future, further tuning of SAM for this type of data could lead to more refined part segmentation.
>
> **For out-of-domain generalization**, we conducted experiments on camouflage object segmentation and provided some visualizations in Figure 17 of the main PDF (rebuttal section, page 24).
>
>
> **Q3:** It could be more convincing if this paper can compare to other SAM-based methods on the same settings and show its advantages. For example, [1] and [2] also work on open-vocabulary segmentation with SAM.
>
> **A3:** Thanks for the question. We first compare the SOTA SAM-based work Frozen Seg (in our Table I, CVPR2024) with SAM-CP because that work experiments on our same setting(COCO->ADE20K). The result indicates that SAM-CP achieves superior open-vocabulary panoptic segmentation outcomes, which verifies the efficacy of composable prompts. Moreover, it demonstrates that the performance is not solely reliant on the internal capacity of SAM:
> |Method|backbone |PQ|AP|mIoU|
> |-|-|-|-|-|
> | Frozen Seg [2] | ConvNext-L | 25.9|16.4|34.4|
> | SAM-CP  | ConvNext-L |27.2|17.0|31.8|
>
> Then, We compared our method with OVSAM on the COCO dataset. Their open-vocabualry setting is differen ty from ours, because the time limition of rebuttal, we compare with them on closed domain at the same setting to show the model ability. We found that OVSAM did not perform as well as our method on both R50 base backbones:
>
> |Method|Detectors|mAP|AP50|
> | - | - | - | - |
> |OVSAM[4]|Faster-RCNN(R50)|35.8|55.6|
> |OVSAM[4]|Detic(Swin-base)|36.7|57.2|
> |SAM-CP(ours)|R50|41.7|59.4|
>
> The comparison between our method and SEEM on the COCO dataset is listed as follows. With the same amount of backbone, we achieved higher performance. In addition to comparing with the several methods you mentioned, we also added a comparison with Semantic SAM. Even though it has point prompts in inference, our method outperforms them by a lot, which fully demonstrates that our method is superior to other SAM-based methods:
> |Method|Backbone|PQ|mAP|mIoU|
> | - | - | - | - | - |
> |SEEM[5]|ViT-L|52.0|43.5|60.2|
> |SAM-CP(ours)|Swin-L|52.7|45.2|60.8|
>
> |Method|Backbone|Prompt(testing)|mIoU|
> | - | - | - | - |
> |Semantic-SAM[6]|Swin-L|point|57.0|
> |Semantic-SAM[6] (reproduce)|Swin-L|point|55.1|
> |SAM-CP(ours)|Swin-L|None|61.8|
>
> [4] Open-Vocabulary SAM. ECCV 2024
>
> [5] Segment Everything Everywhere All at Once. NeurIPS 2023
>
> [6] Semantic-SAM: Segment and Recognize Anything at Any Granularity, arXiv:2307.04767
>
>
> **Q4:** As already discussed and illustrated in the paper, SAM-CP fails at small objects since there are no prompts on small objects. The efficiency is limited by SAM.
>
> **A4:** A good question! We inherit the default option of SAM that starts with placing a fixed grid of point prompts on the image. When the target is small, it is possible that none of the prompts fall on the target, and thus, it is missing. This issue can be alleviated by dynamically adding denser point prompts to the regions with small objects (no re-training is required, but this strategy will slow down the inference). We show an example in Figure 18 of the main PDF (rebuttal section, page 25), where small targets were found with this simple mechanism.
> Figure 19 of the main PDF (rebuttal section, page 25) shows us that SAM is not friendly to small targets only because they occupy a relatively small proportion in the current image and the preset grid points of SAM cannot cover them well. Once we enlarge the small targets, SAM can also automatically segment them well.

---

> ### Author Response · Authors · 2024-11-24
> **Response to Reviewer SuUH (Part # 3)**
>
> **Q5:** SAM can generate fine-grained masks for part, e.g. head, hand, body. Does SAM-CP have the potential to work on open-vocabulary part segmentation? Quantitative experiments and comparisons to related works like HIPIE would enhance the applicability of this work.
>
> **A5:** A good question! It's impressive to see that SAM-CP demonstrates superior open vocabulary capabilities compared to HIPIE on the ADE20K dataset, despite being trained on a smaller set of data. This comparison highlights the potential of SAM-CP's approach in handling diverse and novel categories that may not be present in the training data.
>
> * **Open Vocabulary Comparison on COCO->ADE20K Dataset**: In this setting, SAM-CP was trained on the COCO dataset, while HIPIE was trained on a more extensive collection of datasets, including Object 365, COCO, RefCOCO, Pascal, etc., before being tested on the ADE20K dataset. Despite the smaller training set, SAM-CP outperforms HIPIE in open vocabulary panoptic segmentation on ADE20K. This suggests that SAM-CP may be more effective in generalizing to new scenes and objects.
> |Method|Backbone|PQ|mAP|mIoU|
> | - | - | - | - | - |
> |HIPIE|ViT-H|22.9|19.0|29.0|
> |SAM-CP(ours)|Swin-L|27.2|17.0|31.8|
>
>
> * **Part Segmentation on Pascal Part Dataset:** Because time limitation of rebuttal, we only present the potential of part segmentation without further tuning. For part segmentation, SAM-CP was tested on the Pascal Part dataset, which consists of 4465 images across 93 part categories. The performance metrics are as follows:
> |Part segmentation |mAP|AP$_{50}$|AP$_{75}$|AP$^{s}$|AP$^{m}$|AP$^{l}$|
> |-|-|-|-|-|-|-|
> |SAM-CP|13.7|30.5|10.6|8.3|21.1|24.2|
>
> We also provide some part segmentation visualization in Figure 16 of the main PDF (rebuttal section, page 24).

---

> > ### Comment · Reviewer_SuUH · 2024-11-24
> >
> > Thanks authors for their efforts. After rebuttal, the additional experiments and clarifications have addressed most of my concerns. Therefore, I increased my rating to borderline accept.

---

> > > ### Author Response · Authors · 2024-11-24
> > > **Thank you**
> > >
> > > We thank the reviewer for recognizing our efforts. After the page limit has been confirmed, we will combine the additional content into the final paper.

---

### Official Review · Reviewer_xYh9 · 2024-11-04

**Soundness:** 3
**Presentation:** 2
**Contribution:** 2
**Rating:** 6
**Confidence:** 3

**Summary:**

The paper introduces SAM-CP, which integrates composable prompts for versatile segmentation. It utilizes two levels of queries—semantic and instance—to assign unlabeled patches from SAM with appropriate labels.

In rebuttal, the authors have presented additional experimental results that address many of my concerns. As a result, I am inclined to increase my rating to borderline accept.

**Strengths:**

- The overall architecture designs are well-grounded. The two types of prompts for semantic and instance understanding are intuitively designed to support different segmentation tasks.

- SAM-CP models the two types of prompts with separated queries for better efficiency. The quantitative results on open-vocabulary segmentation also demonstrate the effectiveness of this query-based approach.

- Thorough analysis is provided for the advantages and limitations of the proposed method.

**Weaknesses:**

- The running time might be a potential issue due to the two-level prompt design (especially if an iterative approach is used). A comparison of running time performance with the original SAM and other segmentation models would be informative.

- Experiments: The current baselines do not include models based on SAM. Therefore, it's unclear whether the performance improvements come from the composable prompts or the internal capacity of SAM. Adding baselines building on SAM that perform instance [1] and semantic [2] segmentation can be incorporated to further demonstrate the effectiveness of the proposed composable prompts.

- The figures are low-resolution and notably blurry. Replacing them with high-resolution versions would improve readability and presentation quality.

- The performance of SAM-CP on closed-domain segmentation is not superior over other baselines.

[1] Prompt Segment Anything (https://github.com/RockeyCoss/Prompt-Segment-Anything)
[2] Semantic Segment Anything (https://github.com/fudan-zvg/Semantic-Segment-Anything)

**Questions:**

The design of SAM-CP relies solely on SAM's final predictions, disregarding intermediate features from SAM that could potentially enhance the decoder's performance. It would be valuable to include discussions on this aspect and to investigate whether integrating SAM's intermediate features with the proposed decoder could yield improved results.

---

> ### Author Response · Authors · 2024-11-24
> **Response to Reviewer xYh9 (Part # 1)**
>
> **Q1:** The running time might be a potential issue due to the two-level prompt design (especially if an iterative approach is used). A comparison of running time performance with the original SAM and other segmentation models would be informative.
>
> **A1:** A good question! We opted for two state-of-the-art (SOTA) methods, namely FrozenSeg (SAM-based) and FCCLIP (a non-SAM-Based) , to compare their time performance within our open-vocabulary setting. All experiments were carried out using an RTX 4090.
> The outcomes are as follows:
> |Method|SAM-based|backbone |time of SAM (s/img)|time of non-SAM (s/img)|totol time (s/img)|PQ|AP|mIoU|
> |-|-|-|-|-|-|-|-|-|
> | FCCLIP [3]| no | ConvNext-L | -|0.24| 0.24|25.2|16.4|32.6|
> | Frozen Seg [2]| yes | ConvNext-L | 3.41 | 1.07|4.48|25.9|16.4|34.4|
> | SAM-CP | yes | ConvNext-L | 3.41| 0.21| 3.62|27.2|17.0|31.8|
> | SAM2-CP | yes | ConvNext-L | 1.61| 0.18| 1.79|27.9|18.0|32.6|
>
> **FrozenSeg**, being a SAM-based approach, also requires everything-mode segmentation and thus demands the same SAM time as ours. However, our non-SAM module consumes less time and exhibits better performance. **FCCLIP** (the performance here is a reproduced version by both FrozenSeg and us, which is lower than that in the original paper) is a non-SAM-based method, while SAM-CP demonstrates better segmentation performance.
> In **SAM2-CP**, we substitute SAM with SAM2[2] in SAM-CP. SAM2 is faster than SAM. We utilize the CP trained on SAM-CP and transfer the weights to SAM2-CP without re-train, with the performance being 27.9 PQ. This indicates that our method is modular and can be extended to different patch generation methods (whether stronger or faster). Although the current SAM-based method is more time-consuming than the non-SAM-based method, with the improvement of the segmentation accuracy and effectiveness of the SAM-based foundational model, the new bottom-up perception paradigm designed based on its high generalization ability is valuable, which is also one of our main motivations.
>
> [1] SAM 2: Segment Anything in Images and Videos.
>
> [2] Frozenseg: Harmonizing frozen foundation models for open-vocabulary segmentation. In CVPR 2024.
>
> [3] Convolutions die hard: Open-vocabulary segmentation with single frozen convolutional CLIP. In NeurIPS, 2023.
>
> **Q2:** Experiments: The current baselines do not include models based on SAM. Therefore, it's unclear whether the performance improvements come from the composable prompts or the internal capacity of SAM. Adding baselines building on SAM that perform instance *[1]* and semantic *[2]* segmentation can be incorporated to further demonstrate the effectiveness of the proposed composable prompts.
>
> **A2:** Thanks for the question. We first compare the SOTA SAM-based work Frozen Seg (in our Table I, CVPR2024) with SAM-CP because that work experiments on our same setting (COCO->ADE20K). The result indicates that SAM-CP achieves superior open-vocabulary panoptic segmentation outcomes, which verifies the efficacy of composable prompts. Moreover, it demonstrates that the performance is not solely reliant on the internal capacity of SAM.
>
> |Method|backbone |PQ|AP|mIoU|
> |-|-|-|-|-|
> | Frozen Seg | ConvNext-L | 25.9|16.4|34.4|
> | SAM-CP  | ConvNext-L |27.2|17.0|31.8|
>
> Since Prompt Segment Anything and Semantic Segment Anything are merely engineering projects available on GitHub and lack specific papers for support, our comparison with them is solely based on the datasets and settings utilized in these projects:
> **Instance Segmentation**: In the Prompt Segment Anything project, the segmentation performance is derived from the pretrained-detector H-Deformable-DETR, with the detection bounding-box results employed as prompts for segmentation. It is evident that our method outperforms Prompt Segment Anything in the Res-50 scenario. However, in the case of the Swin-L backbone, due to the SAM patches not being tuned on COCO, SAM-CP might be marginally inferior. But this is also an advantage in that when there is no additional tuning, we can achieve a composite result just by combining the results of the visual foundational model. The instance segmentation performance details are presented as follows:
>
> |Method|Backbone |Prompt generation|Mask AP|
> | - | - | - | - |
> |prompt segment anything|Res50|H-Deformable-DETR+tricks|41.5|
> |prompt segment anything|Swin-L|H-Deformable-DETR+tricks|46.3|
> |SAM-CP(ours)|Res50|None|41.7|
> |SAM-CP(ours)|Swin-L|None|45.2|
>
> **Semantic segmentation:** In the Semantic Segment Anything project, only its performance on the ADE20k dataset has been reported. We adjusted our method to the same settings for comparison. It's worth mentioning that, as mentioned above, our method has no extra tuning, we would like to propose a new bottom-up perception paradigm of "foundation model+ composable prompts" .
>
> |Method|Backbone|mIoU|
> | - | - | - |
> |Semantic segment anything|MIT-B5|47.1|
> |SAM-CP(ours)|Res-50|42.4|
> |SAM-CP(ours)|Swin-L|49.4|

---

> ### Author Response · Authors · 2024-11-24
> **Response to Reviewer xYh9 (Part # 2)**
>
> **Q3:** The figures are low-resolution and notably blurry. Replacing them with high-resolution versions would improve readability and presentation quality.
>
> **A3:** Thanks for the kind reminder. We will change the figures to a higher-resolution version to make them clearer.
>
> **Q4:** The performance of SAM-CP on closed-domain segmentation is not superior over other baselines.
>
> **A4:** Thanks for the question. In the realm of open-vocabulary tasks, SAM-CP has achieved state-of-the-art (SOTA) performance. However, in the closed-vocabulary scenario, our SAM-CP exhibited relatively lower performance in closed-set benchmark testing. The following table, which presents statistically analyzed data from the COCO dataset, delves into the reasons behind this. Primarily, this is because we chose not to fine-tune the masks of the visual foundation model SAM, unlike other methods, such as Mask DINO, which were trained on COCO:
>
> | Method         | mIoU ↑ | mIoU > 0.5 ↑ | MR$_{0.25}$ ↓ | MR$_{0.5}$ ↓ | MR$_{0.75}$ ↓ |
> |----------------|---------|--------------|----------|---------|----------|
> | Mask DINO      | 76.3    | 83.0         | 4.2%     | 10.1%   | 32.3%    |
> | SAM            | 71.1    | 79.5         | 8.9%     | 16.7%   | 39.3%    |
> | SAM+Merging    | 73.3    | 81.2         | 9.0%     | 15.0%   | 33.3%    |
>
> *(The table compares the mean Intersection over Union (mIoU) and missing rates with respect to different IoUs for COCO (val2017) instance segmentation. Here, mIoU refers to the IoU between the highest-IoU proposal and the ground truth, and mIoU${>0.5}$ indicates that we calculate the mIoU only for instances where the best proposal has an IoU higher than 0.5. MR${x}$ represents the proportion of instances without a matched proposal exceeding an IoU of $x$.)*
>
> The table reveals that SAM patches have a higher missing rate compared to the fine-tuned Mask DINO on COCO. Our SAM-CP, which is based on SAM patches without any fine-tuning, thus faces limitations in close-domain performance. However, when SAM-CP merges patches, it achieves better mIoU and lower missing rates, especially MR$_{0.75}$ (39.3% -> 33.3%). This indicates that after applying composable prompts, SAM-CP delivers improved segmentation outcomes. We hope that with the improvement of the segmentation accuracy and effectiveness of the SAM-based foundational model, the new bottom-up perception paradigm designed based on its high generalization ability is valuable, which is also one of our main motivations.
>
>
> **Q5** The design of SAM-CP relies solely on SAM's final predictions, disregarding intermediate features from SAM that could potentially enhance the decoder's performance. It would be valuable to include discussions on this aspect and to investigate whether integrating SAM's intermediate features with the proposed decoder could yield improved results.
>
>
> **A5** A good question! We've observed a significant decrease in performance when attempting to directly integrate SAM's intermediate features with our proposed decoder. Due to the time constraints of the rebuttal period, we plan to delve deeper into this issue post-rebuttal. For now, let's provide some analysis:
>
> As depicted in Figure 15 of the main PDF (rebuttal section, page 24), a t-SNE comparison between SAM and SAM-CP reveals that SAM's features are class-agnostic. SAM predominantly focuses on low-level details such as color and boundary, which are insufficient for the model to distinguish between different categories. Consequently, a straightforward integration of SAM's intermediate features with the proposed decoder is not effective. Our approach of utilizing CLIP for classification and SAM for segmentation embodies the concept of a Mixture of Experts (MoE). However, it's worth investigating whether SAM's features could assist in calculating affinities and facilitate patch merging, which could potentially enhance the overall performance of our model.

---

> ### Author Response · Authors · 2024-11-28
> **Happy for further discussion**
>
> We are sincerely grateful for your thoughtful and comprehensive review of our paper. We have attentively dealt with each of the concerns you raised and highly esteem your valuable constructive feedback, as it has been instrumental in enhancing our work. As the Author-Reviewer discussion period is nearing its conclusion, please feel free to share any additional concerns or suggestions—we would be more than happy to address them. Thank you.

---

> ### Author Response · Authors · 2024-12-01
> **Hope to get a reply and happy to have a further discussion**
>
> This message is because there is only one day left for the Author-Reviewer discussion period, yet we have not received your response. After our rebuttal, the other two reviewers raised their scores, and all other reviewers leaned towards acceptance, which indicates that our response has resolved their issues. However, we have not received your reply. We understand that this may be due to your busy schedule, but we would like to know if we have addressed your concerns. We are also very willing to fully discuss with you to polish our paper. This is a friendly message. Thanks again.

---

### Author Response · Authors · 2024-11-24

Thanks to the reviewers for reading our paper and sharing valuable comments with us.

In this paper, we presented a versatile segmentation approach, SAM-CP, that designs two composable prompts upon the vision foundation model, SAM. By composing the prompts, one can perform semantic, instance, and panoptic segmentation. We developed a sophisticated and efficient training method and showed the ability of SAM-CP in both open-domain and closed-domain segmentation tasks.

In the original review, reviewers are generally content with the submission. The strengths are summarized as follows.
1. The overall architecture designs are well-grounded (**xYh9**). The paper is well-written (**pKRW**)
2.  SAM-CP considers segmentation tasks from a novel perspective(**pKRW**). Transforming the class-agnostic SAM into a usable semantic and instance segmentation is a valuable pursuit(**3KqL**), directly connecting to the question of how to leverage the advance of large-scale vision foundation models into practical application.  This task is important(**3KqL**).
3.  The method is is easy to follow(**SuUH**), with many intuitive figures, tabels and visualizations(**SuUH**, **pKRW**, **3KqL**). The two types of prompts for semantic and instance understanding are intuitively designed (**xYh9**) to support different segmentation tasks. The dynamic mechanism offers novel design insights for handling cross-attention computations(**pKRW**).
4. The quantitative results on open-vocabulary segmentation also demonstrate the effectiveness of this query-based approach(**xYh9**, **SuUH**) and achieve state-of-the-art performance (**3KqL**) under multiple datesets. The overall experiment and results presentation is good(**3KqL**). The ablation studies are sufficient and well explain the design choice of loss functions(**SuUH**).
5. Thorough analysis is provided for the advantages and limitations of the proposed method.(**xYh9**)


 In the rebuttal to each reviewer, we responded to their concerns with texts and supplementary experiments. The pictures are in the main PDF (rebuttal section, page 24-25).

---

### Author Response · Authors · 2024-12-04
**Summary of the rebuttal and author-reviewer discussion**

Dear PC, SAC, AC, and Reviewers,

Thanks again for your time in handling our submission. During the past weeks, we had thorough discussions with the reviewers and they also gave us valuable feedback that will further improve our work. After the rebuttal:

* Reviewer **pKRW** said that *most of his/her concerns have been resolved* and decided to keep the original rating (6).
* Reviewers **SuUH** said that *the additional experiments and clarifications have addressed most of his/her concerns* and decided to raise the score to 6.
* Reviewer **3KqL** said that *the detailed response* has *addressed most of his/her concerns* and decided to raise the score to 6. Specifically, he/she was satisfied with our approach in (1) *generalizing across different granularities when the model is jointly trained on part-level and general datasets* and (2) *finer segmentation can be achieved by interactively using two types of prompts in combination with SAM*.

In summary, all three reviewers who responded after the rebuttal **converged to a score of 6**. It is a pity that Reviewer **xYh9** did not respond to our rebuttal. Some of his/her questions were also raised by other reviewers and have already been dealt with and acknowledged by other reviewers during the rebuttal process.  Should he/she have further comments during the AC-reviewer discussion phase, we are more than happy to accept the suggestions and revise our paper accordingly.

We sincerely thank you for your time and advice and look forward to sharing our work with the community.

Best,

Authors

---

### Meta-Review · Area_Chair_SHCS · 2024-12-23

**Metareview:**

This work presents a simple approach (SAM-CP) to establish two types of composable prompts beyond SAM and then compose them for versatile segmentation.

**Additional Comments On Reviewer Discussion:**

This work has four reviewers, and all of them are positive to accept it. All reviewers think that the rebuttal well addresses their concerns and then keep the positive score or raise their scores to accept it. Hence, this work can be accepted.

---

### Decision · Program_Chairs · 2025-01-22

Accept (Poster)